# Exploring the Potential Use of Near-Miss Information to Improve Construction Safety Performance

**Zhipeng Zhou [1,*], Chaozhi Li [2], Chuanmin Mi [1] and Lingfei Qian [1]**

[1] Department of Management Science and Engineering, College of Economics and Management, Nanjing University of Aeronautics and Astronautics, Nanjing 210000, China; cmmi@nuaa.edu.cn (C.M.); qianlingfei@nuaa.edu.cn (L.Q.)
[2] Nanjing Building Safety Supervision Station, Nanjing 210000, China; lcz321_126@126.com
[*] Correspondence: zhouzhipeng@nuaa.edu.cn

**Abstract:** Construction project management usually has a high risk of safety-related accidents. An opportunity to proactively improve safety performance is with near-miss information, which is regarded as free lessons for safety management. The research status and practice; however, presents a lack of comprehensive understanding on what near-miss information means within the context of construction safety management. The objective of this study is to fill in this gap. The main findings enrich the comprehensive understanding of the near-miss definition, the near-miss causation model, and the process of near-miss management. Considering that near-misses are more tacit and obscure than accidents, the process for near-miss management involves eight stages: discovery, reporting, identification, prioritization, causal analysis, solution, dissemination, and evaluation. The first three stages aim to make near-misses explicit. The other five are adopted to better manage near-miss information, compiled in a well-designed near-miss database (NMDB). Finally, a case study was conducted to show how near-miss information can be utilized to assist in construction safety management. The main potential contributions here are twofold. Firstly, corresponding findings provide a knowledge framework of near-miss information for construction safety researchers who can go on to further study near-miss management. Secondly, the proposed framework contributes to the guidance and encouragement of near-miss practices on construction sites.

**Keywords:** near-miss; accident; construction safety; near-miss causation model; near-miss management; near-miss database

## 1. Introduction

Construction project management is always hampered by safety issues involving frequent but relatively small-scale accidents with diverse hazards, such as falling from heights, being struck by objects, collapse, mechanical injury, explosion, fire, poisoning, and electrocution [1,2]. Despite improvements in safety performance for construction projects in the last decade, nonfatal and fatal accidents still ceaselessly happen [3–5]. Thus, there is still a long way to go before attaining the goal of zero accidents or harm in the construction industry.

The construction industry employs approximately 7% of the world's labor force but is responsible for 30–40% of fatal injuries across all industries [6]. Various accident investigations and reports in construction projects have implied that accidents do not occur without any precursors [7,8]. There are dozens of incidents without any injuries or loss prior to fatal accidents. This type of incident is called a

"near-miss", which is usually neglected by workers and managers on-site. According to Murphy's Law, anything that can go wrong will go wrong [9].

$$P_n = 1 - (1 - P)^n \tag{1}$$

The probability theory can be used to explain Murphy's Law, as below. It is assumed that the probability of an event with low probability is $P$ ($0 < P < 1$) in one test. In Equation (1), $P_n$ represents the probability that an event with low probability happens at least one time after $n$ tests. When $n$ approaches infinity, $P_n$ will approach 1. This indicates that if there are near-misses on construction sites, they will definitely develop into accidents with serious consequences sooner or later. Errors are often viewed differently from accidents. To err is human [10], which implies that it is impossible to completely eliminate human errors. However, accidents can be avoided with timely error detection and correction. Accidents and near-misses have similar causation models. The only difference between them is the consequence due to opportunity factors which belong to random variation [11]. Opportunity factors are beyond normal control and also decide the consequences [12]. If opportunity factors exist, an accident will happen with serious consequences. In contrast, if opportunity factors do not exist, a near-miss will happen with minor consequences. The number of near-misses is often larger than the number of accidents, since opportunity factors do not always exist. Because of the large quantity and similar causation factors, learning from near-misses is a proactive way to prevent accidents from happening and enhance safety performance for construction projects.

The method of near-miss management has been proposed and implemented across a range of fields with high accident risk, such as the aerospace industry [13–15], natural catastrophe [16,17], coal mining [18], the petrochemical industry [19–22], fire service [23], medical care [24,25], sports [26], transportation [27–31], and nuclear power [32]. There are other terms similar to "near-miss", such as near accident, near incident, close call, near hit, non-injured accident, near collision, sentinel event, and warning event. The application of near-miss information has been accepted as an important practice for the prevention of accidents [33,34], because a large pool of near-miss incidents can be collected and analyzed [8,20]. Jones et al. [35] considered near-misses as significant warning precursors to accidents. Incident reporting can become proactive and predictive through capturing near-misses [2,36].

Individual errors of workers or unsafe conditions in construction projects cannot be completely prevented. However, an early warning mechanism can be conducted to help construction project managers create a proactive and effective method for safety management. Workers on project sites should be encouraged to report near-misses that occur during the process of their work. The academia in the field of construction project management has recently started to pay more attention to near-miss management for more effective safety methods. Goldenhar et al. [37] studied the relationships between a variety of job stressors from construction laborers and near-miss outcomes. Saurin et al. [38] constructed a safety planning and control (SPC) model, and near-miss reporting was regarded as a necessary part in this model. Wu et al. [39] regarded near-miss as a type of accident precursor. An investigative model of precursors and immediate contributory factors (PaICFs) was designed on the basis of near-miss incident reporting. Then, a Zigbee RFID (radio frequency identification) sensor network was applied to the autonomous real-time tracking system for near-misses [40]. Cambraia et al. [41] proposed guidelines for identifying, analyzing, and disseminating the information of near-miss incidents on construction sites. The guidelines were tested in a healthcare building project. In order to overcome challenges in collecting near-miss information, an automatic method was proposed for detecting and documenting near-miss falls on the basis of workers' kinematic data captured from wearable inertial measurement units (WIMUs) [42]. Another similar study focused on struck-by accidents in construction projects [43]. Zhou et al. [8] utilized the complex network theory to explore the characteristics of near-miss time series and the mechanism underlying near-misses. These near-miss data were gained from Wuhan subway projects in China.

Although near-miss management has been widely used as an effective tool for safety management in diverse areas, the features of the sectors are not well-integrated. Work processes on construction sites are often loosely defined, unlike the well-defined procedures of other sectors, such as aviation, nuclear, and chemical plants [1]. The features of the complex, dynamic, and unpredictable construction activities and environments, combined with high production pressures and workloads, create a high likelihood of errors [1,3,44]. A survey in the project of Nanjing Subway showed that merely 13.3% of the respondents were familiar with near-miss management. Others only knew a little or did not know at all. The research and practice status reveal that there is an absence of comprehensive knowledge about what near-miss management means within the context of construction projects. This study tries to enhance people's knowledge from three aspects: the near-miss definition, the near-miss causation model, and the processes of near-miss management. We believe it will prove to be beneficial in guiding and encouraging near-miss management research and practice in construction projects.

## 2. Methodology

The main value of near-miss information in the area of safety management is more or less relative to the ability of accident prevention in a proactive way [27,28]. To let practitioners and academics recognize the core value of near-misses, this study aims to explore the potential use of injury- or fatality-free lessons to enhance safety performance in the construction industry. For this type of exploratory study, we used a hybrid approach that borrows thoughts and opinions from other civilian industries, involving literature study, site interview, database development, and case studies.

A literature study is to read through, analyze, and categorize articles [45,46] for determining the essential attributes of materials pertinent to near-miss information in construction safety. Its distinct difference from other approaches is that it does not directly deal with the object under study, but indirectly accesses information from a variety of literature [47]. Literature materials are the crystallization of wisdom, the ocean of knowledge, and have important values for the development of human society, history, culture, and research scholars [48]. Due to limited research in the area of near-miss management in the construction industry, cross-sector learning was conducted for our literature study. Applications of near-miss management methods in other industries can be replanted to the construction industry by examining its applicability. The research team consisted of researchers from the manufacturing, construction, and information systems.

Site interviewing, as a qualitative approach seeks to describe the meanings of central themes of the subjects [49]. The interviewer can pursue in-depth information around the topic [50]. In order to apprehend the practical utilization of near-miss information or knowledge, as well as to offer an appropriate near-miss management method, a closed, fixed-response site interview where all interviewees are asked the same questions and asked to choose answers from among the same set of alternatives was carried out. This format is beneficial for those not practiced in interviewing [51]. Respondent information and seven questions pertinent to construction near-miss practice were designed for answering.

A near-miss database (NMDB) was developed in a user-friendly way using Microsoft Access 2010 [52]. In NMDB, seven classes of objects (including table, query, form, report, page, macro, and module) are designed. Buttons are provided to constitute tools for easy and convenient operation based on Visual Basic for Application (VBA). Among them, tables are the most important objects in a database, as data of other objects are all from tables. This means data from tables are original. Designing tables in a database should conform to two principles [53]. One of these is the information classification principle, which indicates that one table is only pertinent to one subject, and there should not be repetitious information in one table or among tables. Another is the normal forms (NF) principle. There are six normal forms, including first normal form (1NF), second normal form (2NF), third normal form (3NF), Boyce–Codd normal form (BCNF), fourth normal form (4NF), fifth normal form (5NF), domain/key normal form (DKNF), and sixth normal form (6NF) [54–56]. In a general way, the first three normal forms should be fulfilled.

Although case studies remain a controversial approach to data collection, they are widely recognized in many social science studies, especially when in-depth explanations of a social behavior are being sought after [57,58]. A case study is a research approach involving an up-close, in-depth, and detailed examination of the subject of a case, as well as its related contextual conditions [59]. Case studies usually use unstructured interviews or observations to understand the experience or behavior of individuals. The approach of our case study here was contributive to the illustration on how to utilize the main findings of the near-miss definition, near-miss causation model, and the eight stages of near-miss management to assist in the construction safety practice. The results are sure to guide and inspire future study and practice in construction near-miss management.

Figure 1 shows the main processes, which began with a review of the literature pertinent to near-miss management in terms of its technical specifications and applications in different sectors, including the construction sector. The process was further involved with the theoretical research of the near-miss definition and near-miss causation model. A comparative study between two cases of near-misses was conducted to obtain a complete definition of near-miss. Based on the two classes' hazard theory (TCHT), a near-miss causation model was developed to illustrate the interrelationships between near-miss, accidents, incidents, and hazards within the context of the construction site. Considering the tacit feature of near-miss information, an eight-stage framework that relates the effectiveness of a construction company's near-miss management system to its operational and strategic value was derived from a systematic analysis of such events. The purpose of the first three stages was to make near-misses from tacit to explicit, and the other five stages were adopted to better govern near-miss information compiled in a well-designed near-miss database (NMDB). A case study was then empirically implemented in the project of Nanjing Subway Line Four to show how near-miss information could be employed for construction safety management. Finally, a discussion was conducted to present the main findings of this study and the remaining issues of near-miss information management in the construction industry.

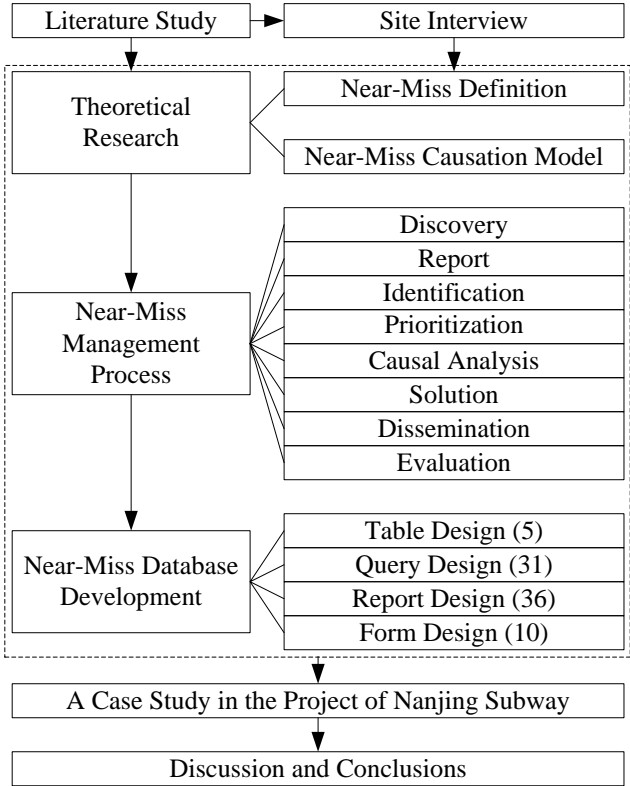

**Figure 1.** The main process for our research.

## 3. Theoretical Research on Near-Misses within Construction Projects

### 3.1. Definition

As its name suggests, a near-miss can be seen as being composed of two parts. One part is "near", which means "close" (e.g., being close to success or failure). The other part is "miss", which means "lose" (e.g., losing a chance to reach a successful goal or being lucky to avoid failure) [53]. Two cases of near-misses are used to illustrate the explanation in Table 1. One case is about a job application, and the other one is about a struck-by incident. Near-misses can be categorized into two types of near success and near failure, according to the analysis in Table 1. Considering that the purpose of this study is to utilize near-miss information for the improvement of safety performance of construction projects, near-misses here are limited to the latter type—an incident close to a failure or an accident.

**Table 1.** Two cases of near-misses.

| Name | A Case of Job Application | A Case of Struck-By Incident |
|---|---|---|
| Description | A man applied for a position. He passed the resume selection, written examination, and two rounds of interviews. However, he made one minor mistake in the last round of interviews conducted by the manager. As a result, he failed to get the position. | There was a heap of material on the scaffolds at a construction project site. A worker walking on the scaffolds knocked the material down accidentally. Part of the material fell down to the ground. Fortunately, there was no one working below the scaffold and nobody was struck by the material. |
| Analysis | This excellent applicant almost got the job. However, he missed at the end because of an unexpected condition and poor performance in the last interview. | If the workers on the ground were closer, a struck-by accident could have happened, and some workers might have been injured or dead. |
| Type | Near success | Near failure |

The apprehension of near-miss is different among various industries. The definition of near-miss needs to be combined with industrial characteristics. A study in the petrochemical industry proposed that the given definition of near-miss should be complete and easy to understand, in order to effectively implement a near-miss management system [35]. Near-misses cannot merely be considered as a type of incident that has potential to cause serious consequences, or identified as an unsafe condition and behavior. The key instrument of the definition should focus on how to promote safety performance. Thus, it was defined as "one type of dangerous state or unsafe act", in the light of incentives and easy comprehension. If there was no disruption, the dangerous state or unsafe act would develop into an accident. Ritwik [12] thought a near-miss offered a chance to improve safety practices. He defined it as an incident or unsafe condition with potential for injury or property damage, including safety-barrier challenges, minor property loss, potential property damage, neighbor complaints, unsafe acts/behaviors, minor release to the environment, unsafe conditions, potential release to the environment, potential injury, and minor injury. In the field of clinical medicine, a near-miss is defined as an event that could lead to negative consequences. Near-misses could provide the opportunity to proactively learn from free lessons. Best et al. [60] defined a near-miss incident as an "unplanned or unforeseen event that could have resulted in human death or other adverse consequences". Another study focused on children's injuries at home. A near-miss is something that happened to your child that could have resulted in him/her being hurt, but fortunately did not [61]. Muermann and Oktem [62] considered near-misses as weak signals, containing the genetic signature of a serious and adverse effect. It was defined as an event, a sequence of events, or an observation of unusual occurrences that possessed the potential of improving a system's operability by reducing the risk of upsets. Some near-misses could eventually cause serious damage. In the Seveso II Directive [63], a near-miss was defined as an unsafe state or act which, if not disrupted, could lead to accidents. These definitions from various sectors indicate that the definition of near-miss does not aim to report near-misses, but to focus on how to employ near-miss information to promote safety performance. Site workers from construction projects are often the main forces of near-miss discovery. Considering the efficiency of

near-miss management and the knowledge level of site workers, easy understanding and a complete definition of near-miss is necessary.

On the basis of the analysis above, the definition of near-miss in safety management of construction projects is given below: Near-misses are one type of incidents with the potential to engender accidents. They do not develop into accidents due to the lack of opportunity factors, which are necessary instruments of accidents in the process of construction project management. A near-miss can be regarded as a type of knowledge about engineering failure, and it is a good opportunity for removing accident risk and promoting safety performance of construction projects. Near-misses consist of unsafe behavior, unsafe conditions, incidents with property loss, incidents with possible damage to the environment, incidents with potential to have more damage, and incidents with a challenged baseline. Figure 2 illustrates the definition of near-miss and depicts the relationship between a near-miss and an accident.

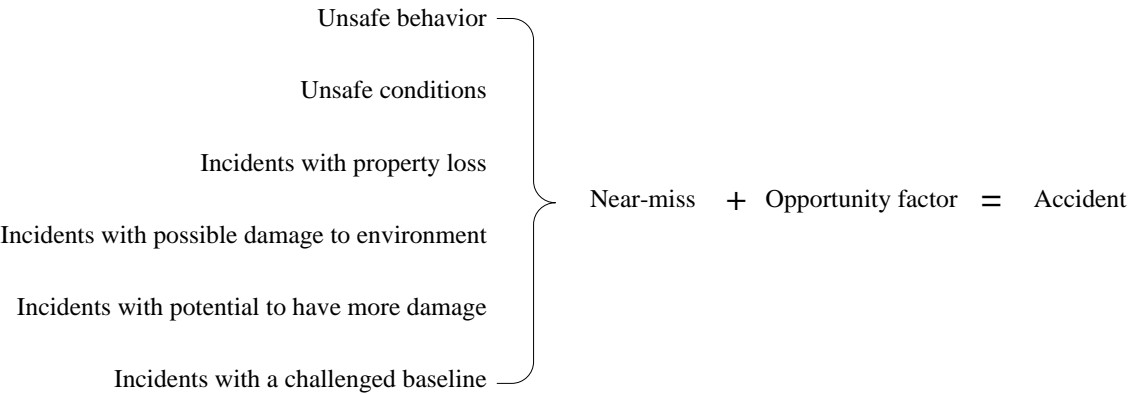

**Figure 2.** Definition of near-miss.

*3.2. Near-Miss Causation Model*

The definition of near-miss in Section 3.1 shows that the causation factors of near-misses are similar to the ones of accidents. It is also verified in other studies [64–67]. The opportunity factor is the only difference between near-misses and accidents. The existence of the opportunity factor will lead to accidents with serious consequences. The causation of both near-misses and accidents can be attributed to unsafe acts or unsafe conditions, which may result in death, injury, loss of property, or damage to the environment. Unsafe acts/conditions are regarded as the source of danger and are called safety hazards. A hazard is a section, zone, place, piece of equipment, or situation with dormant energy and/or material that possesses the risk of injury, death, loss of property, and damage to environment [46]. Based on the energy release theory [68], Chen [69] proposed a theory called the "two classes' hazard theory" (TCHT). Compared with traditional linear incident causation models, the TCHT is more effective at identifying safety hazards in the complex construction project environment of multi-party, multi-trade, and multi-level contracting [2]. Using this theory can better show the impact of safety hazards in the process of near-misses or accidents. Hazards can be categorized into two classes: first-class hazards and second-class hazards.

First-class hazards involve hazardous material or energy with the potential to release in a system, such as a charged conductor or a running vehicle. Table 2 lists the main types of accidents in the process of construction project management and the corresponding hazards categorized as a first-class hazard. A second-class hazard involves unsafe factors that can cause the failure of protective measures for controlling hazardous material or energy release. Causation factors of second-class hazards can be analyzed from three hierarchies: direct, managerial, and basic factors (see Table 3). A first-class hazard, as the carrier of an incident, determines how serious the incident will be. The second-class hazard, as the requisite for an incident, determines the probability of the incident. The second-class hazard is often based upon the first-class hazard. The occurrence of an incident requires both classes

of hazards. A coupling reaction between the two classes of hazards will cause an incident, which may lead to fatality, injury, loss of property, damage to the environment, or loss of a third party. If any of the aforementioned consequences occurs, the incident will be deemed as an accident. If none of them happens, the incident will be deemed as a near-miss. The details of the near-miss causation model are depicted in Figure 3.

**Table 2.** Types of accidents and corresponding hazards categorized as a first-class hazard.

| Type of Accident | Generation and Storage of Energy Source or Hazardous Material | The First-Class Hazard |
|---|---|---|
| Asphyxia | Equipment or material where smoke may engender | Smoke |
| Collapse | Slope of earth and rock engineering, stacking, stock bin, structure, etc. | Soil or rock, material, load, etc. |
| Drowning | River, lake, sea, pond, flood, etc. | Water |
| Electric shock | Power supply | Charged conductor, zone of high step voltage |
| Explosion | Explosive material, combustible gas | Explosive material, combustible gas |
| Fall | Site of large elevation difference, lifting device for staff | Human body |
| Fire | Combustible material | Flame |
| Mechanical injury | Mechanical driving device | Moving part of machinery |
| Struck-by | Equipment, site and operation which engenders dropping, throwing, fracturing, and scattering of objects | Dropping object, throwing object, fracturing object, and scattering object |
| Toxicosis | Equipment, container, and site where toxic and harmful materials engender, store, and cumulate | Toxic and harmful material |
| Vehicle injury | Traction system or ramp which makes vehicle move | Running vehicle |

**Table 3.** Causation factors of second-class hazards.

| Hierarchy | | Causation Factor |
|---|---|---|
| Direct hierarchy | man factor | Behavior, attitude, competency, knowledge, experience, motivation, communication, health, etc. |
| | machine factor | Machine maintenance conditions, quality of machine, protective fittings of machine, suitability of machine, etc. |
| | material factor | Quality of material, quantity of material, suitability of material, material storage conditions, etc. |
| | environmental factor | Site layout, lighting, humidity, temperature, noise, geological conditions, hydrological conditions, etc. |
| Managerial hierarchy | | Safety culture, safety climate, risk management, supervision, training, safety investment, client requirements, etc. |
| Basic hierarchy | | National habits, religious customs and traditions, laws and regulations, economic climate, historical factors, social system, etc. |

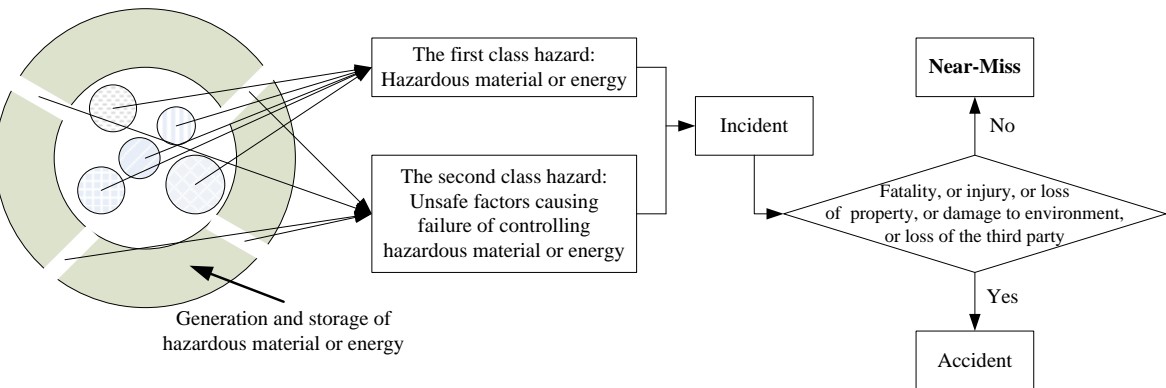

**Figure 3.** Near-miss causation model.

## 4. The Process of Near-Miss Management for Construction Projects

Knowledge is either tacit or explicit [70]. Tacit knowledge is often known by an individual, and it is difficult to transmit tacit knowledge to others in an organization. In contrast, explicit knowledge can be aggregated, coded, and stored. It can also be conveniently transmitted to others. Analogously, accidents with serious consequences are obvious and they receive more attention than near-misses. Near-misses with no injuries are often inconspicuous and prone to being ignored. They can be deemed as the tacit safety knowledge of construction project management. If we would like to make better use of near-misses to assist in construction project safety management, they must firstly be made explicit. Information from near-misses can then be easily utilized.

Considering that construction projects are complex, dynamic, loosely defined processes, and include unpredictable construction tasks and environments [1,71], the process of near-miss management is composed of eight stages: discovery, reporting, identification, prioritization, causal analysis, solution, dissemination, and evaluation, combined with the characteristics of construction projects. The first three stages aim to make the knowledge of near-misses explicit. The next five stages aim at effectively managing and using the explicit near-miss knowledge. Table 4 explores the objective, obstacles in practice, and suggestions for overcoming the obstacles in each stage.

**Table 4.** Eight stages of near-miss management.

| Name | Objective | Obstacle in Practice | Suggestion for Overcoming Obstacles |
|---|---|---|---|
| Discovery | Immediately and completely discover potential near-misses from construction projects. | Know little about near-miss concept and knowledge; lack of consciousness to discover near-misses. | Near-miss definition should be complete and easily understood by site workers from construction projects; train the workers on near-miss discovery. |
| Report | Timely report the near-misses which are discovered in stage one. | Worry about penalty after reporting; workers are not familiar with reporting process or the process is complex and time-consuming; they do not think that near-miss is useful for safety. | Relevant regulations should be proposed for the rewards and punishments about near-miss reporting; near-miss reporting process should be simplified and convenient for workers; the importance of near-misses can be disseminated widely. |
| Identification | Identify whether there are near-misses in the existing database, similar to the reported ones. If yes, the solution can be found from the database. | Near-misses in the database are deficient; neither managers nor experts are sufficiently knowledgeable to identify a near-miss. | Continually improve and update the near-miss database; relevant personnel need to promote the capability to identify near-misses. |
| Prioritization | Analyze the priority of near-misses based on the modified prioritization model. Filter near-misses with high priority. | The guidelines for prioritization are deficient. | Understand the time limit of near-miss transmission; continually improve the modified prioritization model; construct an automatic prioritization system to further efficiency and veracity. |
| Causal Analysis | Determine direct reasons, indirect reasons, and hazards of near-misses. | Insufficient staff to analyze near-misses; information distortion in the process of transmission. | Reporters can assist in the near-miss investigation; adopt causal analysis methods which are often used for accident analysis; invite safety experts to help. |
| Solution | Find solutions according to direct and indirect reasons; select appropriate solutions and implement them. | How to select the appropriate solution from multiple choices; solutions may bring new safety risks. | Consider solutions from multiple aspects of cost, maneuverability, time, and new safety risks. |
| Dissemination | Disseminate the result of near-miss management; make feedback to the initial reporter. | It is not effective, and many workers do not learn from near-misses; it is excessive and results in additional worry. | Use e-learning tools; broadcast near-miss video for workers periodically; based on the characteristics of workers' entertainment, produce squeezers with near-misses for workers. |
| Evaluation | Evaluate the whole process, especially stage six. | How to choose indicators to evaluate. | Compare its actual effectiveness as opposed to its predicted result (e.g., whether the opportunity factor for potential accidents is eliminated). |

The detailed process of near-miss management is displayed in Figure 4. Near-miss discovery involves four potential ways (inclusive) of analyzing near-misses based on existing accident databases, analyzing near-misses based on safety standards or regulations, observing near-misses from site work, and exploring near-misses from construction simulation or virtual reality. A discovered near-miss has limited value unless it is reported and analyzed with proper measures to avoid its recurrence or potential accidents [30,65]. Due to some obstacles shown in Table 4, there is no assurance that every discovered near-miss can be reported. Therefore, the aim of the reporting stage is to ensure that all discovered near-misses are reported from the general workers to construction project managers on-site. A reported near-miss will be checked whether there are similar near-misses in the existing database (the database was developed prior to the implementation of near-miss management; it is used to store

the cases of near-misses). If yes, corresponding solutions to the near-miss can be directly searched from the database. If not, this reported near-miss will be regarded as a new near-miss.

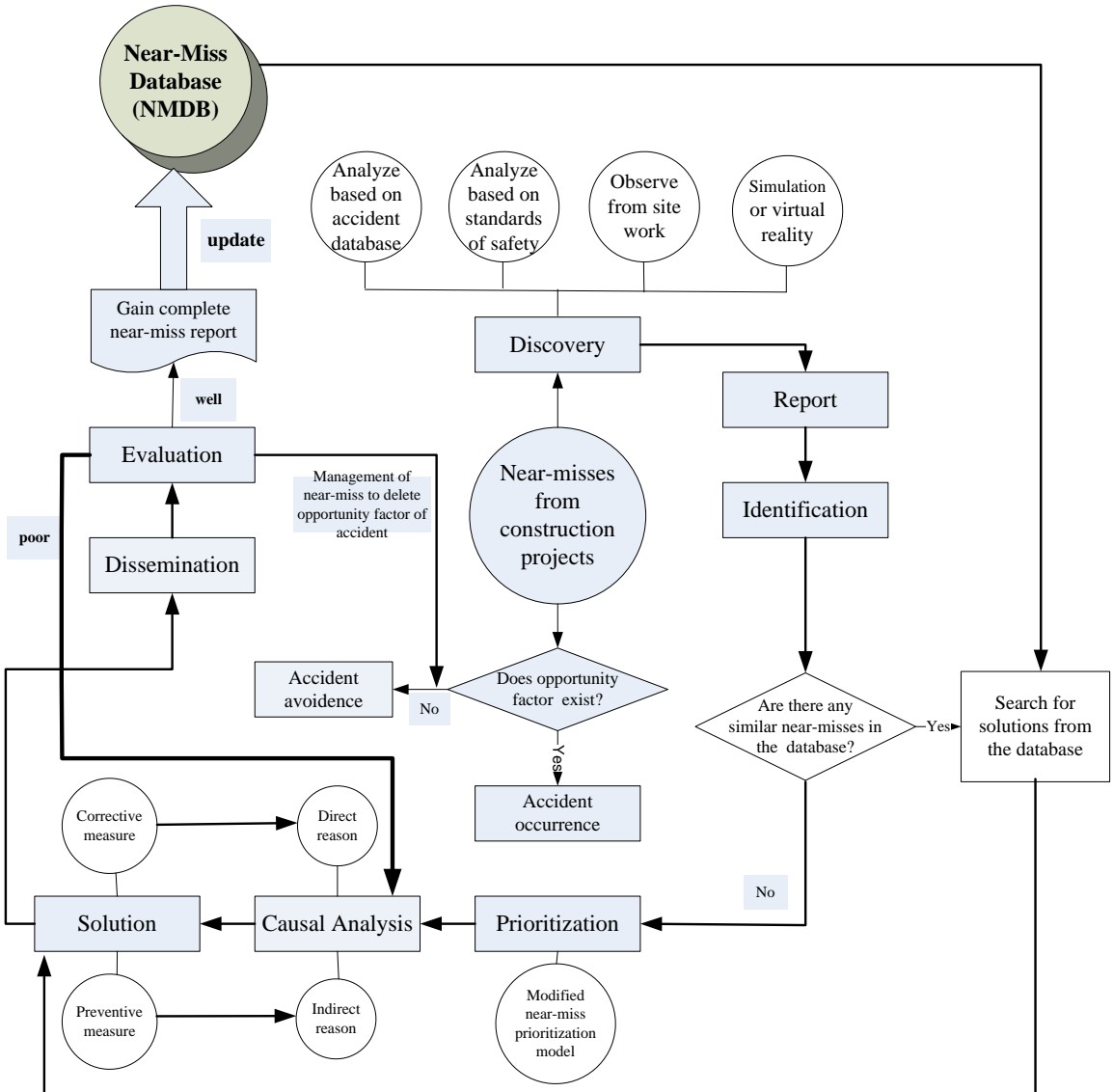

**Figure 4.** The process of near-miss management.

After making near-misses explicit through the first three stages, the next stage is to prioritize near-misses. Limited resources, such as limited time and staff, prevent the simultaneous analysis of all near-misses, but some near-misses with high priority require timely analysis to remove safety risks as soon as possible [72,73]. For example, near-misses have potential for a major accident with serious consequences. Another example is that several similar near-misses are reported simultaneously by different workers. Therefore, safety officers, experts, and managers need to prioritize the near-misses before conducting near-miss analysis. A near-miss decision matrix was proposed by Ritwik [12] to assess the priority of near-misses in the petrochemical industry. Considering the features of the construction industry, a modified near-miss prioritization model can be developed from the consequences of a potential accident (C), near-miss possibility (PO), near-miss proximity (PR), and near-miss learning value (LV). Table 5 illustrates the variables and corresponding weights. The value of near-miss prioritization ($V_{nmp}$) can be calculated by Equation (2). The next stage of casual analysis is to analyze the direct and indirect causations of the near-miss with higher prioritization

firstly [74]. The corresponding corrective and preventive measures will be proposed and implemented, respectively. The result of the near-miss needs to be disseminated and feedback made to the initial reporter. Finally, the performance of the analysis process should be evaluated for its actual effectiveness as opposed to its predicted result (e.g., the elimination of opportunity factors for accident occurrence). On the other hand, the effectiveness of preventive measures is not evident in a short time. A periodic evaluation is required (e.g., to determine if near-miss reports have exhibited a different frequency of occurrence with similar causation factors). If the result is good, a complete near-miss report is gained and adopted to update the near-miss database. It can serve as a potential solution for similar near-misses which may arise in the future, or on other construction project sites. If the result is poor, it is necessary to go back to stage six to conduct causal analysis again to gain other effective measures.

$$V_{nmp} = C \times (PO + PR + LV) \tag{2}$$

In the eight-stage framework, three types of people participate in the process of near-miss management. They are supervisors from the government, researchers from universities or institutions, and employees from construction projects, such as safety managers, safety experts, safety officers, and general workers. Safety managers, safety experts, safety officers, and general workers play key roles in near-miss management as they are stakeholders and direct executors. Table 6 shows the roles which are undertaken by the supervisor, researcher, safety manager, safety expert, safety officer, and general worker. In the process of near-miss management, the main tasks of supervisors are to monitor construction project sites, spread the effectiveness of near-miss information in the organization, and subsidize near-miss management. Researchers pay more attention on studying near-misses and assist in the activities of others in the whole process. The safety manager, safety expert, safety officer, and general worker have different tasks in every stage, as presented in Table 6.

**Table 5.** Variables in the modified near-miss prioritization model.

| Variable | | | Weight |
|---|---|---|---|
| Consequence of potential accident (this variable is considered on the basis of the near-miss causation model in Section 2) | human body | fatality | 3 |
| | | serious injury | 2 |
| | | injury | 1 |
| | property loss | $250,000 and over | 3 |
| | | $50,000 to $250,000 | 2 |
| | | under $50,000 | 1 |
| | damage to environment | high | 3 |
| | | low | 2 |
| | | none | 1 |
| Near-miss possibility (how frequently does the near-miss occur?) | | high | 3 |
| | | general | 2 |
| | | low | 1 |
| Near-miss proximity (how far away is the near-miss from a potential accident?) | | one step | 3 |
| | | two steps | 2 |
| | | more than two steps | 1 |
| | | remote | 0 |
| Near-miss learning value (how useful is the near-miss?) | | useful for the entire company | 3 |
| | | useful for the project | 2 |
| | | useful for the sub-project | 1 |
| | | none | 0 |

**Table 6.** Roles in the process of near-miss management.

| Process of Near-Miss Management | Discovery | Report | Identification | Prioritization |
|---|---|---|---|---|
| supervisor | | supervise, spread, subsidize | | |
| researcher | | study, assist | | |
| safety manager | encourage | encourage | identify | prioritize |
| safety expert | train | train | identify | prioritize |
| safety officer | encourage | unclutter | participate | participate |
| general worker | discover | report | participate | participate |
| **Process of Near-Miss Management** | **Causal Analysis** | **Solution** | **Dissemination** | **Evaluation** |
| supervisor | | supervise, spread, subsidize | | |
| researcher | | study, assist | | |
| safety manager | analyze | analyze | manage | manage |
| safety expert | analyze | analyze | disseminate | evaluate |
| safety officer | participate | implement | implement | participate |
| general worker | assist | implement | implement | participate |

## 5. Developing a Near-Miss Database (NMDB)

### 5.1. Table Design in NMDB

According to the analyses in the Methodology section, five tables, including the near-miss scenario, causal analysis and corresponding solutions, other information, near-miss reporting information, and reporter information, were designed in NMDS (see Tables 7–11). Each table had several fields, and the first field in each table was set as the primary key. Considering the confidentiality of individual and project information, near-miss and reporting information was stored in different tables. Figure 5 illustrates the relationships among the five tables in NMDB.

**Table 7.** Near-miss scenario.

| Field Name | Near-Miss ID | Description | Potential Accident | Task Type | Element Worked on | Material Involved in the Near-Miss | Equipment Involved in the Near-Miss |
|---|---|---|---|---|---|---|---|
| Data Type | text | memo | text | text | text | text | text |
| Field Name | actor number | work platform | environment noise | environment wind | environment lightness | environment temperature | |
| Data Type | number | text | text | number | text | number | |

**Table 8.** Casual analysis and corresponding solutions.

| Field Name | Near-Miss ID | Direct Reasons | Indirect Reasons | Corrective Measures | Preventive Measures |
|---|---|---|---|---|---|
| Data Type | text | memo | memo | memo | memo |

**Table 9.** Other information.

| Field Name | Near-Miss ID | Source | Consequence of Potential Accident | Near-Miss Possibility | Near-Miss Proximity | Near-Miss Learning Value | Risk Degree of Near-Miss |
|---|---|---|---|---|---|---|---|
| Data Type | text | text | number | number | number | number | number |

**Table 10.** Near-miss reporting information.

| Field Name | Near-Miss ID | Reporting Time | Project Name | Project Type | Project Stage | Reporter ID |
|---|---|---|---|---|---|---|
| Data Type | text | date/time | text | text | text | text |

**Table 11.** Reporter information.

| Field Name | Reporter ID | Reporter Name | Gender | Job Position |
|---|---|---|---|---|
| Data Type | text | text | text | text |
| Field Name | education | near-miss training | phone | remark |
| Data Type | text | yes/no | text | memo |

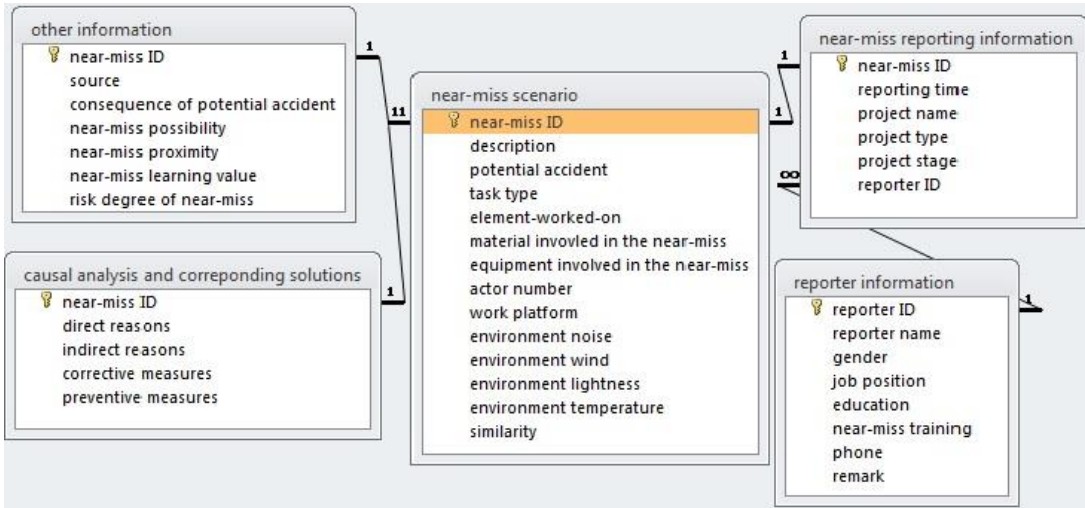

**Figure 5.** Relationships between the five tables in the near-miss database (NMDB).

## 5.2. Query Design in NMDB

Queries are deemed as convenient tools to gain particular data of tables in a database. According to the conditions given, required records from tables can be quickly found by query. These records can then constitute a new table for further processing. Based on the fields in the table of the near-miss scenario, 13 types of queries about scenario information of near-miss were designed. Similarly, the other 18 queries about causal analysis and corresponding solutions, other information, near-miss reporting information, and reporter information were designed.

## 5.3. Report Design in NMDB

Reporting is an effective way to display data from database by format of print. According to the 31 queries in Section 5.2, corresponding reports were set up. Reports of complete records in five tables were also devised. Therefore, there were a total of 36 reports provided in NMDB.

## 5.4. Form Design in NMDB

The form acts as a link between the database and end users. End users can input, search, and manage data conveniently using forms. According to the design of tables, queries, and reports in NMDB, ten forms, including the entrance form, main form, management form of a near-miss case, near-miss identification form, similar near-misses form, management form of near-miss reporter, query form, report form, database instruction form, and exit form, were devised. Due to the length limitation, these forms could not be displayed one by one. Figure 6 shows the main form, which is the most significant form in NMDB. In the main form, the core part is a tab control involving four pages: Case, Identify, Reporter, and Help. On each page, there are three or four commands which act as links between the main form and other forms. In these forms, corresponding Macros and VBA (Visual Basic for Applications) codes are edited to make the tab control and commands more effective.

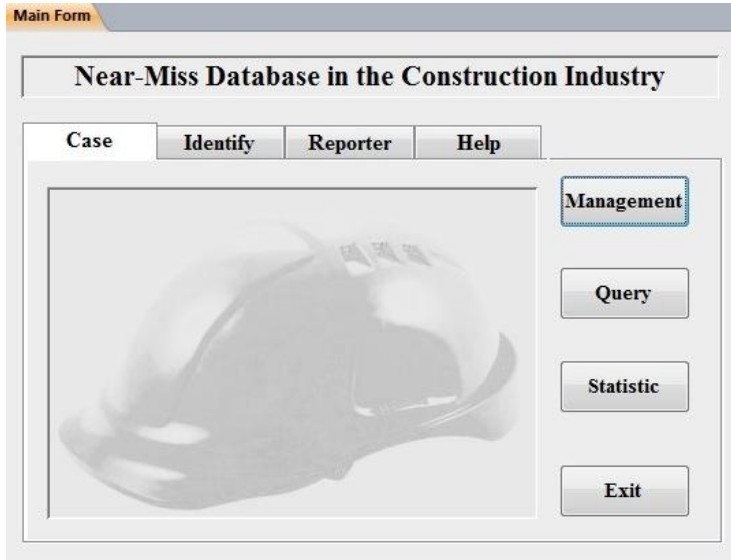

**Figure 6.** Main form in the near-miss database (NMDB).

## 6. A Case Study in the Project of Nanjing Subway

In order to understand the situation of near-miss management of construction projects and make better use of near-miss information, a survey including respondent information and seven pertinent questions was conducted. 72 questionnaires were sent to project managers, safety managers, safety experts, safety officers, general workers, and other staff. A total of 30 questionnaires were effectively completed and returned. The effective response rate was 41.7%. The results are shown in Table 12.

**Table 12.** Questionnaire and results.

| | Respondent Information | | | Question | |
|---|---|---|---|---|---|
| Age | 20 or younger than 20 | 6.7% | Q1: How long have you worked in the construction industry? | 1 year or below | 13.3% |
| | 21–30 | 20% | | 2–5 years | 70% |
| | 31–40 | 40% | | 6–10 years | 13.3% |
| | 41–50 | 33.3% | | 11–15 years | 3.3% |
| | 50 or older than 50 | 0 | | 16 years or above | 0 |
| Education background | Middle school or lower | 23.3% | Q2: Have you ever been involved in a construction accident? | No | 30% |
| | High school | 50% | | Yes, but the accident is not serious | 50% |
| | Junior college | 16.7% | | Yes, the accident is serious | 20% |
| | Bachelor's degree | 10% | Q3: Do you know about near-miss management? | No | 23.3% |
| | Master's degree | 0 | | Yes, I know a little about that | 63.3% |
| | Doctoral degree | 0 | | Yes, I am familiar with that | 13.3% |
| Role of respondent's company | Employer | 3.3% | Q4: Has near-miss management been applied in the project before? | No | 66.7% |
| | Contractor | 86.7% | | Yes | 33.3% |
| | Supervisor | 6.7% | Q5: Do you think that near-miss management is an effective way to reduce accidents? | No | 16.7% |
| | Designer | 0 | | Yes, a little effective | 30% |
| | Supplier | 3.3% | | Yes, very effective | 53.3% |
| Position of respondent | Project manager | 3.3% | Q6: How many near-misses have there been in the project? | Few | 36.7% |
| | Safety manager | 3.3% | | A few | 40% |
| | Safety expert | 3.3% | | Many | 23.3% |
| | Safety officer | 6.7% | Q7: Would you like to report, when you discover near-misses? | No | 6.7% |
| | General worker | 66.7% | | Maybe | 70% |
| | others | 16.7% | | Yes, absolutely | 23.3% |

Most of the respondents were general workers from the contractor division, according to respondent information. Respondents with less than five years' full-time work experience in construction projects accounted for 83.3%. This implies that the employee turnover rate is high

in the construction industry. A high employee rate may influence the efficiency of near-miss management and increase its managerial difficulty. Only four respondents were familiar with near-miss management—one safety manager, one safety expert, and two safety officers. This indicated that most general workers did not know about near-misses or only knew a little. It will thus be necessary for workers to be instructed and trained about what near-miss information means within the context of the construction industry. More than half of the respondents admitted that near-miss management would be a very effective way to reduce or prevent accidents. However, only seven respondents said that they would like to report them, upon discovering near-misses. The others selected "Maybe" or "No". One respondent said, "Th near-miss reporting process looks complicated and it will take much time to complete this process. If I am not busy, I may report. However, if I am busy, I will not spend additional time reporting near-misses." Another respondent said, "Near-misses usually result from mistakes which are made by ourselves. I may be punished, if managers know that I have made a mistake. I will not report near-misses." These reasons for not reporting near-misses were also found in other studies [2,8,29,75]. Therefore, a set of regulations or rules should be developed to overcome the obstacles and encourage site workers to report near-misses when they discover them on construction project sites.

The process of near-miss management was applied in the project of Nanjing Subway Line Four. This project is 33.8 km long and has 18 stations. At 5:20 p.m. on 17 December 2014, a serious accident happened in the section of TA08 of Nanjing Subway Line Four. A steel frame collapsed in the process of reinforcement work 18 m underground. This accident caused four deaths and three injuries. Table 13 shows the near-misses which were reported in the construction project in one week. There were merely five near-misses in total. A possible reason was that most of the site workers in the project were not familiar with near-miss management.

**Table 13.** Near-misses reported during one week.

| Number | Near-Miss Incident Description |
| --- | --- |
| NM1 | There was one hole on the scaffold. The size of the hole was as large as the workers' feet. A worker was plastering the ceiling of the subway station on the scaffold at that time. He missed his footing at the hole and fell down to the scaffold. He was lucky that he was not injured. |
| NM2 | A long section of a steel bar was bulging from the tunnel wall in the walking path. It hit the safety helmets of workers several times. |
| NM3 | The alarm sound of a subway construction railcar was not loud enough. One day, a worker was working along the track. He didn't hear the alarm sound when the subway construction railcar was coming. The driver shouted to him in time, and the worker left the track immediately, avoiding the vehicle collision accident. |
| NM4 | A worker leaned on the guardrails of scaffolds which were a little loose. The worker almost fell down to the ground. A workmate near him gave him a hand. |
| NM5 | A worker walked backwards to talk about something with his workmate behind him. There were several bricks on the walking area. He missed the bricks, tripped, and fell down. Luckily, he was not hurt. |

According to the process of near-miss management shown in Figure 4, five near-miss incidents were discovered by general workers and reported to relevant safety staff and managers. The next stage was to determine whether there were cases similar to the five near-misses in NMDB. The "Identify Form" can be applied to assist in identification. NM1 was taken as an example in Figure 7. The scenario information of NM1 was filled in the "Identify Form" on the basis of the near-miss reporting form. The similarity calculation between the new near-miss and existing near-misses in NMDB was then conducted. The default threshold value of similarity was 0.8. The result is shown in Figure 8. Three near-misses in NMDB were similar to NM1 in the condition of default threshold. Users can change the threshold and re-calculate the similarity to obtain appropriately similar near-misses. While the value of the threshold decreases, the suitability of the solutions for similar near-misses will definitely diminish. As near-miss 006 has the highest similarity, and its corrective and preventive measures can be considered, such as replacing the broken panel by an intact one, or nailing a cover over the hole.

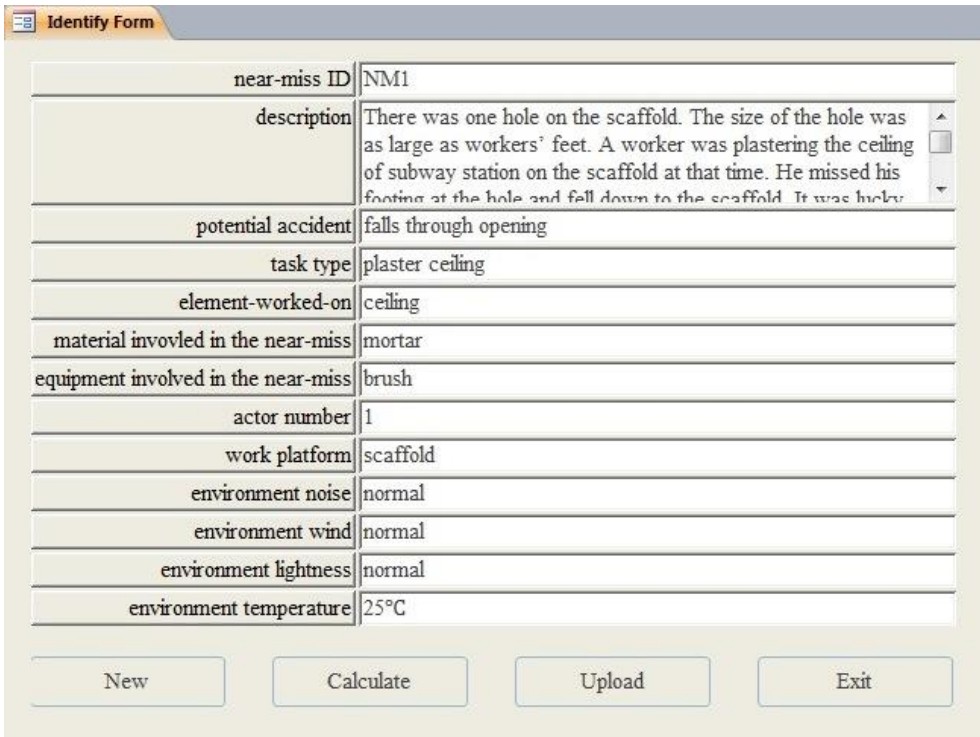

**Figure 7.** Near-Miss 1 (NM1) Identification.

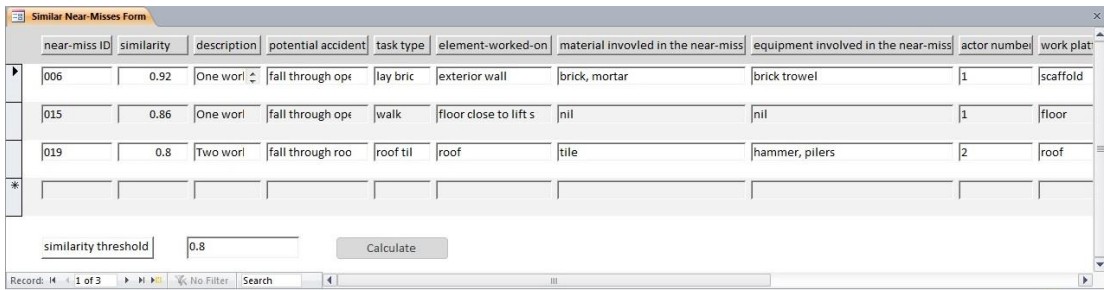

**Figure 8.** Existing near-misses similar to Near-Miss 1 (NM1).

As to NM2, NM3, NM4, and NM5, there were no similar near-misses in NMDB. They were then regarded as new near-misses. According to Figure 4, the prioritization of the four near-misses should be conducted before analyzing causal factors and proposing measures. The four near-miss incidents were screened based on the modified near-miss prioritization model. A potential accident of NM2 is a struck-by accident which might result in serious injury. A potential accident of NM3 is a vehicle collision accident which might result in fatality. A potential accident of NM4 is a fall from a scaffolds accident which might result in fatality. As to NM5, a potential accident is a trip accident which might result in injury. The index of potential accident consequence for each near-miss is filled in Table 14. Other indices of near-miss possibility, proximity, and learning value were also assessed, as shown Table 14. As a result, NM3 and NM4 had the highest value of near-miss prioritization, followed by NM2 and NM5. NM3 and NM4 were firstly analyzed in the next stages. Table 15 explores the cause analysis and corresponding solutions of NM3 and NM4. NM3 and NM4 were introduced and shared with other workers in regular safety meetings after the implementation of solutions. Rewards were given to the reporters in order to encourage them to actively participate in near-miss management. After implementing the corrective and preventive measures in Table 15, the corresponding opportunity factors were removed, and similar near-misses did not take place on

the construction site. This indicated that the result of near-miss analysis was positive. A final report file was used to update the near-miss database.

**Table 14.** Prioritization index of near-misses.

| Near-Miss | Consequence of Potential Accident | Near-Miss Possibility | Near-Miss Proximity | Near-Miss Learning Value | $V_{nmp}$ |
|---|---|---|---|---|---|
| NM2 | 2 | 3 | 2 | 2 | 14 |
| NM3 | 3 | 3 | 3 | 2 | 24 |
| NM4 | 3 | 2 | 3 | 3 | 24 |
| NM5 | 1 | 1 | 0 | 3 | 4 |

**Table 15.** Causal analysis and corresponding solutions.

| Near-Miss | Priority | Causal Analysis | | Solution | |
|---|---|---|---|---|---|
| | | Direct Reason | Indirect Reason | Corrective Measure | Preventive Measure |
| NM3 | 1 | The alarm sound was not loud enough. | The subway construction railcar was not well-maintained. | Repair the alarm equipment in the railcar. | Periodic maintenance of equipment should be guaranteed. Only certified and workers can operate railcar. |
| NM4 | 1 | The guardrails of the scaffold were not firmly fixed. | Guardrails were not well-installed and maintained. | Fix the guardrail on the scaffolds. | Guardrails should be designed and installed by competent workers. Periodic inspection on guardrails should be conducted by safety officers. |
| NM2 | 2 | A long section of steel existed in the walking path. | Correct construction process was violated. | Cut the long section of steel on the tunnel wall. | Correct construction process and rules should be obeyed. |
| NM5 | 3 | Bricks were left on the walking area. | The requirements of site housekeeping were not met. | Remove the bricks from the walk area. | Site housekeeping should comply with particular requirements. |

NM2 was then analyzed in the same way. The direct reason for NM2 was that a long section of steel existed in the walking path. The indirect reason of NM2 was the violation of construction processes. A corresponding measure was adopted to address the near-miss. A red cloth was used to wrap the long section of steel, so that it could warn site workers when they walked around it. NM2 was also introduced and shared with other workers in the regular safety meetings after the implementation of solutions. However, site workers were not alerted to the long section of steel with red cloth, and similar near-misses reoccurred. This indicated that the result of the evaluation was poor, and it would be necessary to analyze NM2 again. Another solution of cutting the long section of steel was adopted. After that, no one was hit by the steel. This indicated that the result of evaluation was good. A complete reporting file was used to update the near-miss database. Finally, NM5 was analyzed in similar way, and the results are shown in Table 15.

## 7. Discussion

According to Maslow's hierarchy of needs [76,77], safety is the second fundamental need of human beings, and there is no exception for the employees in the construction industry. Near-misses have been considered as a good opportunity to promote safety performance in various sectors, such as the petrochemical [22], fire protection [23], medical care [24], transportation [28], and nuclear power industries [32]. Research status and practice presents a lack of comprehensive knowledge about what near-miss information means within the context of construction project management. The main findings of this study add to the knowledge from three aspects: the near-miss definition, the near-miss causation model, and the near-miss management process.

Although previous studies focusing on near-miss management in the construction industry provides various definitions of near-miss [8,40,41], the feature that construction activity processes are usually not standardized [78,79], compared to other industries such as manufacturing and aviation, was neglected. The proposed broad definition of the near-miss concept here involves six categories: unsafe behavior, unsafe conditions, incidents with property loss, incidents with possible damage to the environment, incidents with potential to have more damage, and incidents with a challenged baseline, making it easily understood and discovered by construction workers.

A near-miss causation model developed on the basis of TCHT [69] clearly illustrates the nature of near-miss, including the interrelationships and differences between a near-miss and an accident. It is argued that one person incurs injury or death through a change of energy. Near-misses or accidents are often the result of the insufficient control of energy. The advantage of the proposed model is that it reveals the nature of near-miss occurrence from an ontological perspective. It is not a blame model, but tries to understand how and why a near-miss arises in such a situation. This model regards near-misses as a control problem, which is beneficial to hazard estimation, injury causation analysis, and energy transference control. Given that the TCHT is inadequate in hierarchical analysis of the second class of hazards, the hierarchy of the causal influences model [80] can be considered to assist in modifying the two classes' hazard model. The proposed near-miss causation model in this study focuses on the transference of unrestrained energy from sources through possible paths, and to potential receivers.

According to knowledge management, near-miss belongs to a type of tacit failure knowledge [2,7]. The framework for near-miss management contains eight stages: discovery, reporting, identification, prioritization, causal analysis, solution, dissemination, and evaluation. This systematic framework provides the feasibility of near-miss management on construction sites. Theoretical and practical contributions are twofold. As to theoretical contribution, corresponding findings provide a knowledge framework of near-miss information for construction safety researchers who can go on to further study regarding near-miss management. As for a practical contribution, the proposed framework contributes to the guidance and encouragement of near-miss practices on construction sites. However, various obstacles will continue to hinder the utilization of near-miss management information for construction projects, as shown in Table 4. Some of the obstacles were found through the survey. Although corresponding suggestions were proposed in this study, obstacles cannot be overcome immediately. Because some obstacles are related to historical factors, the company culture, and individual habit, it will take a long time to overcome them [7,41]. Hence, the obstacles impeding near-miss management should be further studied and tackled. In addition, two more opportunities for future studies can be determined: (1) accident precursors can be explored from the near-miss database to support real-time early warnings on construction project sites; and (2) near-misses can act as leading indicators to evaluate the safety performance of a construction project or a company for a certain period of time.

## 8. Conclusions

This study proposed a framework integrating a literature study, site interview, database development, and case study. Its main purpose was to explore the potential use of near-miss information for promoting safety performance during the process of construction project management. Compared with traditional safety management methods, such as accident-based analysis and safety checklists in the construction industry, assisting safety management by near-miss information has been found to be more powerful for improving safety performance. This study focused on near-miss information to address key injury or fatality risk issues in the construction industry. A complete definition of the near-miss concept and a near-miss causation model made it indigenous to all stakeholders on construction sites and provided necessary flexibility to accommodate differences between various practices. An eight-stage systematic framework offered the opportunity to capture all current or potential safety problems, and to resolve them according to their prioritization, one by one. A case study was implemented in the project of Nanjing Subway Line Four, in order to illustrate how to utilize near-miss as effective information to deal with construction safety-related issues in a proactive way. The proposed framework and corresponding findings can be used for guiding and encouraging future study and practice in near-miss management within the context of construction projects.

However, there are two limitations in this study. Subway construction is merely one sector in the entire construction industry. There are also some other sectors, such as building construction, bridge construction, expressway construction, and dam construction. More case studies should be carried out in other types of construction projects for illustrating the general usability of near-miss information.

The other limitation is the small number of near-misses which were collected in one week. A longer period of time should be taken for collecting much more cases of near-misses, which would be good for partly solving the quantitative problems based on statistical models. Additionally, more training methods and incentives are necessary for construction employees on-site to discover and report as many potential near-misses as possible.

**Author Contributions:** Z.Z. drafted the manuscript, C.M. and L.Q. contributed to research methodology, and C.L. assisted in the process of construction site interview. All authors have read and approved the final version of the manuscript.

**Funding:** National Natural Science Foundation of China (NSFC) (Nos. 51508273 and 71871116), the University Philosophy and Social Science Research Project in Jiangsu Province (No. 2016SJD630001), China Postdoctoral Science Foundation (No. 2016M591846), and Jiangsu Province Postdoctoral Science Foundation (No. 1601188B).

**Acknowledgments:** The authors gratefully acknowledge the respondents from the project of Nanjing Subway Line Four. The anonymous reviewers and the editors of this study are also acknowledged for their constructive comments and suggestions.

**Conflicts of Interest:** The authors declare no conflicts of interest.

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
