# Peer review of "Exploring the Potential Use of Near-Miss Information to Improve Construction Safety Performance"

_sustainability, doi:10.3390/su11051264_

Reviewer 1 Report

The manuscript is about near misses which is an under-researched topic.Below are the comments for the authors to consider:

Clarity of the title needs to be improved. "to further safety performance" seems to be inappropriate. 

Abstract: Structure should be improved, highlight the research problem, aim, key findings, contribution.

Introduction: Research problem is not well defined. Needs substantial improvement. Aims are not derived coherently from the research problems. Current aims do not provide clear directions what to include in the manuscript. 

Research method section is completely missing. 

Sections 2 and 3 seem to be literature review. However, they also seem to play the role of findings. If they are findings, it is unknown how the literature was selected and analysed to come up with the findings. 

Section 4 is about developing a near-miss database. This section is rather disconnected from the previous sections. Does it contribute to achieving any aim and objectives?

Section 5 case study is weak. Why a case study is needed? Again, how does this section contribute to achieving the aim and objectives of the manuscript? 

There is no separate section of Discussion. If Discussion is meant to infuse with the findings, the discussion is minimal and weak. 

Conclusions can be improved by clearly relating the findings back to the aim and objectives and highlight the significance and contributions of the paper. 

Author Response

Comment 1: Clarity of the title needs to be improved. "to further safety performance" seems to be inappropriate.

Response:

According to the comment, the title was modified as below.

Exploring the Potential Use of Near-Miss Information to Improve Construction Safety Performance

Comment 2: Abstract Structure should be improved, highlight the research problem, aim, key findings, contribution.

Response:

Considering the suggestion, the structure of abstract was adjusted to clarify the research problem, aim, key findings and contribution. Please see the new edition of abstract as below.

Construction project management usually has the high risk of safety accidents. An opportunity to proactively improve safety performance is near-miss which is regarded as free lessons for safety management. The research status and practice, however, presents a lack of comprehensive understanding what near-miss information means within the context of construction safety management. The objective of this study is to fill in the gap. Main findings enrich the knowledge inclusive of near-miss definition, near-miss causation model, and process of near-miss management. Considering that near-misses are more tacit and obscure than accidents, the process for near-miss management involves eight stages of discovery, report, identification, prioritization, causal analysis, solution, dissemination and evaluation. The first three stages aim to make near-misses explicit. The other five are adopted to better manage near-miss information compiled in a well-designed near-miss database (NMDB). A case study was finally conducted to show how to utilize near-miss information to assist construction safety management. Main potential contributions here are twofold. Firstly, corresponding findings provide a knowledge framework of near-miss information for construction safety researchers who can go on further study on near-miss management. Secondly, the proposed framework is contributive to the guidance and encourage of near-miss practices on construction sites.

Comment 3: Research problem is not well defined. Needs substantial improvement. Aims are not derived coherently from the research problems. Current aims do not provide clear directions what to include in the manuscript.

Response:

According to the comment, modifications were made to clarify the research problem in both sections of abstract and introduction. The research status and practice presents a lack of comprehensive knowledge understanding about what near-miss information means within the context of construction safety management. In order to fill in this gap, the aim of this study is to obtain a knowledge framework of near-miss information pertinent to construction safety management. Additionally, the section of methodology was supplemented for providing clear directions what to include in the manuscript.

Comment 4: Research method section is completely missing.

Response:

Thank for the suggestion. The section of methodology was supplemented in the manuscript as below.

Main value of near-miss information in the area of safety management is more or less relative to the ability of accident prevention in a proactive way. To let practitioners and academics recognize the core value of near-misses, this study aims to explore the potential use of injury or fatality free lessons to enhance safety performance in the construction industry. A hybrid framework, borrowing thoughts and opinions from other civilian industries, is proposed to guide the exploration in the study.

Figure 1 shows main process which began with a review of literature pertinent to near-miss management in terms of its technical specifications and applications in different sectors including the construction sector. The process further involved the theoretical research of near-miss definition and near-miss causation model. A comparative study between two cases of near-misses was conducted to obtain a complete definition of near-miss. Based on two classes’ hazard theory (TCHT), near-miss causation model was developed to illustrate the interrelationships among near-miss, accident, incident and hazard within the context of construction site. Considering the tacit feature of near-miss information, an eight-stage framework that relates the effectiveness of a construction company’s near-miss management system to its operational and strategic value can be derived from a systematic analysis of such events. The purpose of the first three stages is to make near-misses from tacit to explicit. The other five stages are adopted to better govern near-miss information compiled in a well-designed near-miss database (NMDB). A case study is then empirically implemented to show how to utilize near-miss information for construction safety management. Finally discussion is conducted to present main findings of this study and remaining issues of near-miss information management in the construction industry.

Figure 1. Main Process for this Research

Comment 5: Sections 2 and 3 seem to be literature review. However, they also seem to play the role of findings. If they are findings, it is unknown how the literature was selected and analysed to come up with the findings.

Response:

Section 2 and section 3 play the role of findings based on literature study, and some referrences are cited. Section 2 focuses on near-miss definition and near-miss causation model. The apprehension of near-miss is different among various industries. The definition of near-miss need be combined with industrial characteristics. Hence near-miss definition from other industries such as aerospace industry, natural catastrophe, coal mining, the petrochemical industry, fire service, medical care, and transportation was referred. These definitions from various sectors indicate that the definition of near-miss does not aim to report near-misses, but to focus on how to utilize near-misses to further safety performance. Site workers from construction projects are often the main forces of near-miss discovery. Considering the efficiency of near-miss management and knowledge level of site workers, easy understanding and complete definition of near-miss is necessary. On the basis of the analysis above, the definition of near-miss in safety management of construction projects was derived. Comparing with traditional linear incident causation model, two classes’ hazard theory is more effective to identify safety hazards in the complex construction project environment of multi-party, multi-trade and multi-level contracting. Using this theory can better illustrate the impact of safety hazards in the process of near-misses or accidents. On the basis of the two classes’ hazard theory, near-miss causation model was developed.

Section 3 focuses on the process of near-miss management. Considering that construction projects are complex, dynamic, loosely defined processes, and include unpredictable construction tasks and environments, the process of near-miss management is composed of eight stages of discovery, report, identification, prioritization, causal analysis, solution, dissemination and evaluation, combined with the characteristics of construction projects. The first three stages aim to make the knowledge of near-misses explicit. The next five stages aim to effectively manage and use the explicit near-miss knowledge. The eight stages of near-miss management are based on studies and practices in other industries and incorporate all those important stages. This eight-stage systematic framework offers the opportunity to capture all current or potential safety problems on construction sites, and resolve them according to the prioritization one by one.

Comment 6: Section 4 is about developing a near-miss database. This section is rather disconnected from the previous sections. Does it contribute to achieving any aim and objectives?

Response:

According to Figure 3 in Section 3, the near-miss database is an important part in the process of near-miss management. During the stage of near-miss identification, it is necessary to check whether there are any similar near-misses stored in the database. If yes, corresponding solutions will be found directly from the database. After the completion of all eight stages of near-miss management, a near-miss report will be got to supplement or update existing cases in the database. Hence, this database supports the entire process of near-miss management.

Comment 7: Section 5 case study is weak. Why a case study is needed? Again, how does this section contribute to achieving the aim and objectives of the manuscript?

Response:

The section of case study in this manuscript is contributive to the illustration on how to utilize main findings of near-miss definition, near-miss causation model, and eight stages of near-miss management to assist in construction safety practice. This will guide and encourage future study and practice in near-miss management within the context of construction project management.

Comment 8: There is no separate section of Discussion. If Discussion is meant to infuse with the findings, the discussion is minimal and weak.

Response:

Considering the suggestion, a stand-alone section of discussion was supplemented as Section 7 in the new version of manuscript as below.

According to Maslow's hierarchy of needs, safety is a fundamental physical and psychological need of human beings, and there is no exception for the employees in the construction industry. Near-misses have been considered as a good opportunity to promote safety performance in the petrochemical industry, fire protection, medical care, transportation, nuclear power and so on. The research status and practice presents a lack of comprehensive knowledge about what near-miss means within the context of the construction project management. Main findings of this study add to that knowledge from three aspects of near-miss definition, near-miss causation model, and near-miss management process. The broad definition of the near-miss concept makes it easily understood and discovered by construction workers. A near-miss causation model developed on the basis of two classes’ hazard theory, clearly illustrates the nature of near-miss, the interrelationships and differences between near-miss and accident. According to knowledge management, near-miss belongs to a type of tacit failure knowledge. The framework for near-miss management contains eight stages of discovery, report, identification, prioritization, causal analysis, solution, dissemination and evaluation. This systematic framework provides the feasibility of near-miss management on construction sites. In the project of Nanjing Subway Line Four, a case study was conducted to display how to deploy near-miss as effective information to deal with construction safety-related issues in a proactive way.

However various obstacles will continue to hinder the utilization of near-miss management information for construction projects, as shown in Table 4. Some of the obstacles were found through the survey. Although corresponding suggestions were proposed in this study, obstacles cannot be overcome immediately. Because some obstacles are related to historical factors, company culture and individual habit, it will take a long time to overcome them. Hence, the obstacles impeding near-miss management should be further studied and tackled. In addition, two more opportunities for future studies can be determined: (1) accident precursors can be explored from the near-miss database to support real-time early warning on construction project sites; and (2) near-misses can act as leading indicators to evaluate the safety performance of a construction project or a company for a certain period of time.

Comment 9: Conclusions can be improved by clearly relating the findings back to the aim and objectives and highlight the significance and contributions of the paper.

Response:

According to this comment, the structure of conclusions was adjusted correspondingly as below, and the limitations of this study were added in the end.

This study proposes a framework integrating literature study, site interview, database development and case study. Its main purpose is to explore potential use of near-miss information for promoting safety performance during the process of construction project management. Compared with traditional construction safety management such as accident-based analysis and safety checklist, safety management assisting by near-miss information is a more proactive way to improve safety performance. This study focuses on near-miss information to address key injury or fatality risk issues in the construction industry. A complete definition of the near-miss concept and a near-miss causation model make it indigenous to all stakeholders on construction sites and provides necessary flexibility to accommodate differences between various practices. An eight-stage systematic framework offers the opportunity to capture all current or potential safety problems, and resolve them according to the prioritization one by one. The proposed framework and corresponding findings can be used for guiding and encouraging future study and practice in near-miss management within construction project management.

There are still two limitations for this study. Subway construction is merely one sector in the entire construction industry. There are some other sectors such as building construction, bridge construction, expressway construction, and dam construction. A case study was conducted on subway construction site in order to show how to utilize main findings of near-miss definition, near-miss causation model, and eight stages of near-miss management in practice. More case studies should be carried out in other types of construction projects for illustrating the general usability of near-miss information. The other limitation is the small number of near-misses which were collected in one week. A longer period of time will be taken for collecting much more number of near-misses, which is good for partly solving quantitative problems based on statistical models. Additionally, more training methods and incentives are necessary for construction employees on site to discover and report potential near-misses as many as possible.

Finally two editions of manuscript (one has track changes, and the other one is clean copy) are provided for your better reviewing.

Reviewer 2 Report

Comment for Authors

This study presents the potential use of near-miss information in construction project management. The introduction clearly established the rationale for the study with good evaluation of literature and previous studies. However, the paper has a major problem it has failed to clearly explained the method and methodology used in conducting the investigation. This should be done.

Introduction:

The authors should provide citation to support their statement in line 107 – 110.

In line 223 – 226 the authors should explain how they arrived at the eight stages of near-miss listed. Again in line 230, the authors discuss how they arrived at the detail process of near-miss management shown in Figure 3.

In line 243 the authors are now trying to explain the process proposed but what is not clear to the reviewer is; what is the evidence that informed the proposed near-miss process. The authors should explain how this knowledge was constructed.

Line 243 – 266 the authors should use citations to support their argument.

Table 4 the authors should explain how they arrive at the information presented in the Table. Is this based on empirical study?

Table 5 the authors should explain how they arrived the information presented in the table including the weighting.

Research Method

The major issue with this manuscript is that it lacks clear method and methodology. The author should create methodology section. This should clearly describe and discuss approach used in the current study. At the moment it very difficult to follow how the various information presented were arrived it.

The authors should provide and justify the rationale for the approaches used in the investigation.

In line 352 – 253 the result indicates that 83.3% of the respondent has less than five years’ experience in the construction industry. The authors should explain the implication of this for their findings.

The authors should discuss the result presented in Table 12.

Discussion

The authors have not discussed the findings presented. This should be done.

 Conclusion

The current conclusion looks more like a discussion this should be improved. The authors should make firm conclusion based on their findings.

The authors should also highlight the limitation of the current study.

Author Response

Response to the Comments from Reviewer#2

Comment 1: The authors should provide citation to support their statement in line 107 – 110.

Response:

It is not clear for this comment, because the content of statement from 107 to 110 is shown as in Figure I. There are merely two words in Line 107. Line 108 is empty. Line 109 and 110 are titles and subtitles. If reviewer#2 mentioned Line 111-114, a citation (Zhou et al., 2012) was added in the manuscript.

Zhou, Z., Li, Q., and Wu, W. (2012). “Developing a versatile subway construction incident database (SCID) for the safety management.” Journal of Construction Engineering and Management, 138(10), 1169-1180.

Figure I. A Screenshot from Line 106 to 114

Comment 2: In line 223 – 226 the authors should explain how they arrived at the eight stages of near-miss listed. Again in line 230, the authors discuss how they arrived at the detail process of near-miss management shown in Figure 3.

Response:

Thanks for the suggestion. On one hand, the eight stages of near-miss management are based on studies and practices in other industries and incorporate all those important stages. On the other hand, the eight stages of near-miss management follow the investigation process of a general accident. Comparing to general accidents, near-misses are a bit tacit. Considering this difference between near-miss and accident, the eight stages were separated into two groups. The first three stages (discovery, report and identification) aim to make the information of near-misses explicit. The next five stages (prioritization, causal analysis, solution, dissemination and evaluation) aim to effectively manage and use the explicit near-miss information.

For the detail process of near-miss management, some of stages were directly analyzed by authors, and the others also referred to some research or practice. For example, in the stage of prioritization, a near-miss decision matrix was employed. This matrix was proposed by Ritwik (2002) to assess the priority of near-misses in the petrochemical industry. Considering the characteristics of the construction industry, a modified near-miss prioritization model were developed from consequence of potential accident (C), near-miss possibility (PO), near-miss proximity (PR), and near-miss learning value (LV). Table 5 in the manuscript illustrates the variables and corresponding weights. The value of near-miss prioritization (Vnmp) can be calculated in following equation.

Vnmp = C × (PO+PR+LV)

Ritwik, U. (2002). “Risk-based approach to near miss.” Hydrocarb. Process., 10, 93-96.

Comment 3: In line 243 the authors are now trying to explain the process proposed but what is not clear to the reviewer is; what is the evidence that informed the proposed near-miss process. The authors should explain how this knowledge was constructed.

Response:

Thanks for the suggestion. Construction accidents have the characteristics of specific position, massive workload, underground or outdoor work, liquidity of construction, mono-construction and construction with long period. Combined with those characteristics, the implementation process of near-miss management is divided into 8 stages of discovery, report, identification, prioritization, casual analysis, solution, dissemination and evaluation.

Near-miss discovery involves four potential ways inclusive of analyzing near-misses based on existing accident databases, analyzing near-misses based on safety standards or regulations, observing near-misses from site work, and exploring near-misses from construction simulation or virtual reality. The aim of the reporting stage is to ensure that all discovered near-misses should be reported from general workers to construction project managers on site. A reported near-miss will be checked whether there are similar near-misses in the existing database. If yes, corresponding solutions to the near-miss can be directly searched from the database. If not, this reported near-miss will be regarded as a new near-miss. The stage of prioritization aims to analyze the priority of near-misses based on the modified prioritization model and filter near-misses with high priority. The next stage of casual analysis is to analyze the direct and indirect causations of the near-miss with higher prioritization firstly. Corresponding corrective and preventive measures will be proposed and implemented, respectively. The result of near-miss needs to be disseminated and made feedback to the initial reporter. Finally, the performance of the analysis process should be evaluated for its actual effectiveness as opposed to its predicted result. If the result is well, a complete near-miss report is gained and adopted to update the near-miss database. If the result is poor, it is necessary to go back to stage six to conduct causal analysis again to gain other effective measures.

Comment 4: Line 243 – 266 the authors should use citations to support their argument.

Response:

According to the suggestion, several citations were added as below.

Dillon, R. L., & Tinsley, C. H. (2016). Near-miss events, risk messages, and decision making. Environment Systems & Decisions, 36(1), 34-44.

Baumard, P., & Starbuck, W. H. (2005). Learning from failures: Why it may not happen. Long Range Planning, 38(3), 281-298.

Bier, V. M., & Mosleh, A. (1990). The analysis of accident precursors and near misses: implications for risk assessment and risk management. Reliability Engineering & System Safety, 27(1), 91-101.

Comment 5: Table 4 the authors should explain how they arrive at the information presented in the Table. Is this based on empirical study?

Response:

Thanks for the suggestion. Based on interviews with participating member companies the objectives, obstacles in practice, and suggestions for overcoming obstacles for each stage were obtained.

Comment 6: Table 5 the authors should explain how they arrived the information presented in the table including the weighting.

Response:

The information of the first two variables of consequence of potential accident and near-miss possibility including their weighting was determined based on empirical study. The information of the last two variables of near-miss proximity and near-miss learning value including the weighting was employed from the area of chemical safety management.

Comment 7: The major issue with this manuscript is that it lacks clear method and methodology. The author should create methodology section. This should clearly describe and discuss approach used in the current study. At the moment it very difficult to follow how the various information presented were arrived it. The authors should provide and justify the rationale for the approaches used in the investigation.

Response:

According to the comment, a stand-alone section of methodology was added as below. Figure II was added to show the main process for this research.

Main value of near-miss information in the area of safety management is more or less relative to the ability of accident prevention in a proactive way. To let practitioners and academics recognize the core value of near-misses, this study aims to explore the potential use of injury or fatality free lessons to enhance safety performance in the construction industry. A hybrid framework, borrowing thoughts and opinions from other civilian industries, is proposed to guide the exploration in the study.

Figure 1 shows main process which began with a review of literature pertinent to near-miss management in terms of its technical specifications and applications in different sectors including the construction sector. The process further involved the theoretical research of near-miss definition and near-miss causation model. A comparative study between two cases of near-misses was conducted to obtain a complete definition of near-miss. Based on two classes’ hazard theory (TCHT), near-miss causation model was developed to illustrate the interrelationships among near-miss, accident, incident and hazard within the context of construction site. Considering the tacit feature of near-miss information, an eight-stage framework that relates the effectiveness of a construction company’s near-miss management system to its operational and strategic value can be derived from a systematic analysis of such events. The purpose of the first three stages is to make near-misses from tacit to explicit. The other five stages are adopted to better govern near-miss information compiled in a well-designed near-miss database (NMDB). A case study is then empirically implemented to show how to utilize near-miss information for construction safety management. Finally discussion is conducted to present main findings of this study and remaining issues of near-miss information management in the construction industry.

Figure II. Main Process for this Research

Comment 8: In line 352-253 the result indicates that 83.3% of the respondent has less than five years’ experience in the construction industry. The authors should explain the implication of this for their findings.

Response:

Thanks for the suggestion. This implies that employee turnover rate is high in the construction industry. High employee rate may influence the efficiency of near-miss management and increase its managerial difficulty.

Comment 9: The authors should discuss the result presented in Table 12.

Response:

Thanks for the comment. Line 340-357 was supplemented and adjusted to present the corresponding results of Table 12.

Comment 10: The authors have not discussed the findings presented. This should be done.

Response:

According to the suggestion, a separate discussion was added as section 7 in the new edition of manuscript. The details are as below.

According to Maslow's hierarchy of needs, safety is a fundamental physical and psychological need of human beings, and there is no exception for the employees in the construction industry. Near-misses have been considered as a good opportunity to promote safety performance in the petrochemical industry, fire protection, medical care, transportation, nuclear power and so on. The research status and practice presents a lack of comprehensive knowledge about what near-miss means within the context of the construction project management. Main findings of this study add to that knowledge from three aspects of near-miss definition, near-miss causation model, and near-miss management process. The broad definition of the near-miss concept makes it easily understood and discovered by construction workers. A near-miss causation model developed on the basis of two classes’ hazard theory, clearly illustrates the nature of near-miss, the interrelationships and differences between near-miss and accident. According to knowledge management, near-miss belongs to a type of tacit failure knowledge. The framework for near-miss management contains eight stages of discovery, report, identification, prioritization, causal analysis, solution, dissemination and evaluation. This systematic framework provides the feasibility of near-miss management on construction sites. In the project of Nanjing Subway Line Four, a case study was conducted to display how to deploy near-miss as effective information to deal with construction safety-related issues in a proactive way.

However various obstacles will continue to hinder the utilization of near-miss management information for construction projects, as shown in Table 4. Some of the obstacles were found through the survey. Although corresponding suggestions were proposed in this study, obstacles cannot be overcome immediately. Because some obstacles are related to historical factors, company culture and individual habit, it will take a long time to overcome them. Hence, the obstacles impeding near-miss management should be further studied and tackled. In addition, two more opportunities for future studies can be determined: (1) accident precursors can be explored from the near-miss database to support real-time early warning on construction project sites; and (2) near-misses can act as leading indicators to evaluate the safety performance of a construction project or a company for a certain period of time.

Comment 11: The current conclusion looks more like a discussion this should be improved. The authors should make firm conclusion based on their findings.

Response:

According to this comment, the structure of conclusions was adjusted correspondingly as below.

This study proposes a framework integrating literature study, site interview, database development and case study. Its main purpose is to explore potential use of near-miss information for promoting safety performance during the process of construction project management. Compared with traditional construction safety management such as accident-based analysis and safety checklist, safety management assisting by near-miss information is a more proactive way to improve safety performance. This study focuses on near-miss information to address key injury or fatality risk issues in the construction industry. A compete definition of the near-miss concept and a near-miss causation model make it indigenous to all stakeholders on construction sites and provides necessary flexibility to accommodate differences between various practices. An eight-stage systematic framework offers the opportunity to capture all current or potential safety problems, and resolve them according to the prioritization one by one. The proposed framework and corresponding findings can be used for guiding and encouraging future study and practice in near-miss management within construction project management.

There are still two limitations for this study. Subway construction is merely one sector in the entire construction industry. There are some other sectors such as building construction, bridge construction, expressway construction, and dam construction. A case study was conducted on subway construction site in order to show how to utilize main findings of near-miss definition, near-miss causation model, and eight stages of near-miss management in practice. More case studies should be carried out in other types of construction projects for illustrating the general usability of near-miss information. The other limitation is the small number of near-misses which were collected in one week. A longer period of time will be taken for collecting much more number of near-misses, which is good for partly solving quantitative problems based on statistical models. Additionally, more training methods and incentives are necessary for construction employees on site to discover and report potential near-misses as many as possible.

Comment 12: The authors should also highlight the limitation of the current study.

Response:

Two limitations were supplementd in the section of conclusions. Subway construction is merely one sector in the entire construction industry. There are some other sectors such as building construction, bridge construction, expressway construction, and dam construction. A case study was conducted on subway construction site in order to show how to utilize main findings of near-miss definition, near-miss causation model, and eight stages of near-miss management in practice. More case studies should be carried out in other types of construction projects for illustrating the general usability of near-miss information. The other limitation is the small number of near-misses which were collected in one week. A longer period of time will be taken for collecting much more number of near-misses, which is good for partly solving quantitative problems based on statistical models. Additionally, more training methods and incentives are necessary for construction employees on site to discover and report potential near-misses as many as possible.

Finally two editions of manuscript (one has track changes, and the other one is clean copy) are provided for your better reviewing.

Reviewer 3 Report

The article strives to develop a model based on the near miss information. This is an interesting topic and could be very helpful for the construction industry. 

I have a few suggestions that can improve the article:

1- The affiliations of the authors do not match the number of authors.

2- the abstract needs to inform the result and the significance of the research

3- the limitation of the research and the use of proposed model should be discussed perhaps by allocating a specific section

4- theoretical and practical contribution of the research should be discussed.

5- there are a few editorial mistakes ( e.g in the conclusion there is an error prompt) 

6- the entire paper needs proofreading

Author Response

Response to the Comments from Reviewer#3

Comment 1: The affiliations of the authors do not match the number of authors

Response:

According to the comment, the affiliations of the authors were updated in the manuscript.

Comment 2: The abstract needs to inform the result and the significance of the research.

Response:

Considering the suggestion, the section of abstract was adjusted as below. The results and the significance of the research were clarified accordingly.

Construction project management usually has the high risk of safety accidents. An opportunity to proactively improve safety performance is near-miss which is regarded as free lessons for safety management. The research status and practice, however, presents a lack of comprehensive understanding what near-miss information means within the context of construction safety management. The objective of this study is to fill in the gap. Main findings enrich the knowledge inclusive of near-miss definition, near-miss causation model, and process of near-miss management. Considering that near-misses are more tacit and obscure than accidents, the process for near-miss management involves eight stages of discovery, report, identification, prioritization, causal analysis, solution, dissemination and evaluation. The first three stages aim to make near-misses explicit. The other five are adopted to better manage near-miss information compiled in a well-designed near-miss database (NMDB). A case study was finally conducted to show how to utilize near-miss information to assist construction safety management. Main potential contributions here are twofold. Firstly, corresponding findings provide a knowledge framework of near-miss information for construction safety researchers who can go on further study on near-miss management. Secondly, the proposed framework is contributive to the guidance and encourage of near-miss practices on construction sites.

Comment 3: The limitation of the research and the use of proposed model should be discussed perhaps by allocating a specific section

Response:

According to the comment, a separate section of methodology was provided for discussing the research framework. Additionally, two limitations of this study were supplemented in the section of conclusion as below.

There are still two limitations for this study. Subway construction is merely one sector in the entire construction industry. There are some other sectors such as building construction, bridge construction, expressway construction, and dam construction. A case study was conducted on subway construction site in order to show how to utilize main findings of near-miss definition, near-miss causation model, and eight stages of near-miss management in practice. More case studies should be carried out in other types of construction projects for illustrating the general usability of near-miss information. The other limitation is the small number of near-misses which were collected in one week. A longer period of time will be taken for collecting much more number of near-misses, which is good for partly solving quantitative problems based on statistical models. Additionally, more training methods and incentives are necessary for construction employees on site to discover and report potential near-misses as many as possible.

Comment 4: Theoretical and practical contribution of the research should be discussed.

Response:

According to the suggestion, theoretical and practical contributions of this study were supplemented in the section of discussion as below.

Theoretical and practical contributions here are twofold. As to theoretical contribution, corresponding findings provide a knowledge framework of near-miss information for construction safety researchers who can go on further study on near-miss management. As to practical contribution, the proposed framework is contributive to the guidance and encourage of near-miss practices on construction sites.

Comment 5: There are a few editorial mistakes ( e.g in the conclusion there is an error prompt)

Response:

According to the comment, the entire manuscript was checked to correct editorial mistakes, especially for the section of conclusions. Furthermore, the structure of conclusions was adjusted for better understanding of this study.

Comment 6: The entire paper needs proofreading.

Response:

According to suggestion, the entire paper was proofread.

Finally two editions of manuscript (one has track changes, and the other one is clean copy) are provided for your better reviewing.

Round  2

Reviewer 1 Report

No further comments.

Author Response

Due to no further comments, there are no responses to Reviewer#1.

Thanks Reviewer#1 for the constructive comments and suggestions.

Reviewer 2 Report

The reviewer believed that the authors   have addressed some of the initial concerns raised. However, the method section, the discussion and the conclusion could still be improved.

Method

The author(s) have titled the section methodology which is a good thing to do, but there is no methodological discussion. The authors should use literature to support the approach used in the investigation. Looking through the method section, there is no single citations to previous study to justify the approach used in the current investigation. The authors should explain what was done in the case study. What were the unit of analysis on the case study? How was the evidence collaborated with other evidence? How was the evidence collected? If interview was used how many people were interviewed. All these need to be clearly stated in the

Discussion

The authors should discuss their finding in relation to previous study. They should support the discussion of their result with literature. Going through the discussion section I rarely find any citation.

Conclusion

The author(s) stated that “Its main purpose is to explore potential use of near-miss information for promoting safety performance during the process of construction project management”. I expect the authors to shown this in the conclusion.  For instance was this  achieved based on the evidence from the case study or not. This should be part of the conclusion.

Author Response

Response to the Comments from Reviewer#2

Comment 1: Method: The author(s) have titled the section methodology which is a good thing to do, but there is no methodological discussion. The authors should use literature to support the approach used in the investigation. Looking through the method section, there is no single citations to previous study to justify the approach used in the current investigation. The authors should explain what was done in the case study. What were the unit of analysis on the case study? How was the evidence collaborated with other evidence? How was the evidence collected? If interview was used how many people were interviewed. All these need to be clearly stated in the

Response:

Thanks for the comments and suggestions. The section of methodology were adjusted and supplemented as below. The hybrid approach involving literature study, site interview, database development and case study are analyzed one by one, and corresponding references are cited.

Main value of near-miss information in the area of safety management is more or less relative to the ability of accident prevention in a proactive way (Aldred, 2016; Aldred and Goodman, 2018). To let practitioners and academics recognize the core value of near-misses, this study aims to explore the potential use of injury or fatality free lessons to enhance safety performance in the construction industry. For this type of exploratory study, a hybrid approach borrows thoughts and opinions from other civilian industries, involving literature study, site interview, database development and case study.

Literature study is to read through, analyze and categorize articles (Moons et al., 2019; Zhou et al., 2015) for determining the essential attributes of materials pertinent to near-miss information in construction safety. Its distinct difference from other approaches is that it does not directly deal with the object under study, but to indirectly access to information from a variety of literatures (Sueyoshi et al., 2017). Literature materials are the crystallization of wisdom, are the ocean of knowledge, have important values for the development of human society, history, culture and research scholars (Nakano and Akikawa, 2014). Due to limited research in the area of near-miss management in the construction industry, cross-sector learning was conducted for literature study. Applications of near-miss management methods in other industries can be replanted to the construction industry by examining its applicability. The formulation of the research team has included researchers from manufacturing, construction, and information system.

Site interview as a qualitative approach seeks to describe the meanings of central themes in the life world of the subjects (Kvale, 1996). The interviewer can pursue in-depth information around the topic (Foddy, 1993). In order to apprehend the practical utilization of near-miss information or knowledge and offer an appropriate near-miss management method, a closed, fixed-response site interview where all interviewees are asked the same questions and asked to choose answers from among the same set of alternatives are carried out. This format is beneficial for those not practiced in interviewing (Azarpazhooh et al., 2008). Respondent information and seven questions pertinent to construction near-miss practice are designed for answering.

A near-miss database (NMDB) is developed in a user friendly way using Microsoft Access 2010 (Microsoft Corporation, 2010). In NMDB, seven classes of objects (including Table, Query, Form, Report, Page, Macro and Module) are designed. Buttons are provided to constitute tools for easy and convenient operation based on Visual Basic for Application (VBA). Among them, tables are the most important objects in a database as data of other objects are all from tables. It means data from tables are original. Designing tables in a database should conform to two principles (Zhou et al., 2012). One principle is information classification principle, which indicates that one table is only pertinent to one subject and there should not be repetitious information in one table or among tables. Another principle is normal forms (NF) principle. There are six normal forms, including First Normal Form(1NF), Second Normal Form (2NF), Third Normal Form (3NF), Boyce-Codd Normal Form(BCNF), Fourth Normal Form(4NF), Fifth Normal Form(5NF), Domain/Key Normal Form(DKNF), Sixth Normal Form(6NF) (Codd, 1970; Fagin, 1981; Kent, 1983). In a general way, the first three normal forms should be fulfilled.

Although case study remains a controversial approach to data collection, they are widely recognized in many social science studies especially when in-depth explanations of a social behavior are sought after (Ridder, 2017; Thomas, 2011). A case study is a research approach involving an up-close, in-depth, and detailed examination of a subject of the case, as well as its related contextual conditions (Yin, 2013). Case studies usually use unstructured interviews or observations to understand the experience or behavior of individuals. The approach of case study here is contributive to the illustration on how to utilize main findings of near-miss definition, near-miss causation model, and eight stages of near-miss management to assist in construction safety practice. The results will guide and inspire future study and practice in construction near-miss management.

Figure 1 shows main process which began with a review of literature pertinent to near-miss management in terms of its technical specifications and applications in different sectors including the construction sector. The process further involved the theoretical research of near-miss definition and near-miss causation model. A comparative study between two cases of near-misses was conducted to obtain a complete definition of near-miss. Based on two classes’ hazard theory (TCHT), near-miss causation model was developed to illustrate the interrelationships among near-miss, accident, incident and hazard within the context of construction site. Considering the tacit feature of near-miss information, an eight-stage framework that relates the effectiveness of a construction company’s near-miss management system to its operational and strategic value can be derived from a systematic analysis of such events. The purpose of the first three stages is to make near-misses from tacit to explicit. The other five stages are adopted to better govern near-miss information compiled in a well-designed near-miss database (NMDB). A case study is then empirically implemented in the project of Nanjing Subway Line Four to show how to employ near-miss information for construction safety management. Finally discussion is conducted to present main findings of this study and remaining issues of near-miss information management in the construction industry.

Aldred, R. (2016). “Cycling near misses: Their frequency, impact, and prevention.” Transport. Res. A-Pol., 90, 69-83.

Aldred, R., and Goodman, A. (2018). “Predictors of the frequency and subjective experience of cycling near misses: Findings from the first two years of the UK near miss project.” Accident Analysis and Preventi., 110, 161-170.

Azarpazhooh, A., Ryding, W. H., and Leake, J. L. (2008). “Structured or unstructured personnel interviews?” Healthcare Management Forum, 21(4), 33-43.

Codd, E. F. (1970). “A relational model of data for large shared data banks.” Communications of the ACM, 13(6), 377-387.

Fagin, R. (1981). “A normal form for relational databases that is based on domains and keys.” ACM Transactions on Database Systems (TODS), 6(3), 387-415.

Foddy, W. (1993). Constructing questions for interviews. Cambridge University Press, Cambridge, UK.

Kent, W. (1983). “A simple guide to five normal forms in relational database theory. Communications of the ACM, 26(2), 120-125.

Kvale, S. (1996). Interviews: An introduction to qualitative research interviewing. Sage Publications, Thousand Oaks, USA.

Microsoft Corporation (MS), 2010. Microsoft Access 2010, Washington, USA.

Moons, K., Waeyenbergh, G., and Pintelon, L. (2019). “Measuring the logistics performance of internal hospital supply chains – A literature study.” Omega, 82, 205-217.

Nakano, M., and Akikawa, T. (2014). “Literature review of empirical studies on SCM using the SSPP paradigm.” Int. J. Prod. Econ., 153, 35-45.

Ridder, H. (2017). “The theory contribution of case study research designs.” Business Research, 10, 281-305.

Sueyoshi, T., Yuan, Y., and Goto, M. (2017). “A literature study for DEA applied to energy and environment.” Energ. Econ., 62, 104-124.

Thomas, G. (2011). How to do your case study: A guide for students and researchers. Sage Publications, Thousand Oaks, USA.

Yin, R. K. (2013). Case study research: Design and methods (5th ed.). Sage Publications, Thousand Oaks, USA.

Zhou, Z., Goh, Y. M., and Li, Q. (2015). “Overview and analysis of safety management studies in the construction industry.” Safety Sci., 72, 337-350.

Zhou, Z., Li, Q., and Wu, W. (2012). “Developing a versatile subway construction incident database (SCID) for the safety management.” J. of Constr. Eng. Manage., 138(10), 1169-1180.

Comment 2: Discussion: The authors should discuss their finding in relation to previous study. They should support the discussion of their result with literature. Going through the discussion section I rarely find any citation.

Response:

According to the comments, the structure of discussion section was modified. The authors discussed the findings in relation to previous studies from the construction industry or other industries. Accordingly, some references were cited in the section of discussion. The modified discussion was provided as below.

According to Maslow's hierarchy of needs (Maslow, 1943; Maslow, 1954), safety is the second fundamental need of human beings, and there is no exception for the employees in the construction industry. Near-misses have been considered as a good opportunity to promote safety performance in various sectors such as petrochemical industry (Kongsvik et al., 2012), fire protection (Taylor et al., 2014), medical care (Baig et al., 2018), transportation (Aldred and Goodman, 2018), and nuclear power (Uth and Wiese, 2004). The research status and practice presents a lack of comprehensive knowledge about what near-miss information means within the context of the construction project management. Main findings of this study add to the knowledge from three aspects of near-miss definition, near-miss causation model, and near-miss management process.

Although previous studies focusing on near-miss management in the construction industry provides various definitions of near-miss (Cambraia et al., 2010; Wu et al., 2010b; Zhou et al., 2017), the feature that construction activity processes are usually not standardized (Golzarpoor et al., 2018; Oliveira et al., 2018) comparing to other industries such as manufacturing, and aviation was neglected. The proposed broad definition of the near-miss concept here involves six categorizes of unsafe behavior, unsafe conditions, incidents with property loss, incidents with possible damage to the environment, incidents with potential to have more damage and incidents with a challenged baseline, making it easily understood and discovered by construction workers.

A near-miss causation model developed on the basis of two classes’ hazard theory (Chen, 1996), clearly illustrates the nature of near-miss, the interrelationships and differences between near-miss and accident. It is argued that one person incurs injury or death through a change of energy. Near-misses or accidents are often the result of the insufficient control of energy. The advantage of the proposed model is that it reveals the nature of near-miss occurrence from an ontological perspective. It is not a blame model, but tries to understand how and why a near-miss arises in such a situation. This model regards near-misses as a control problem, which is beneficial to hazard estimation, injury causation analysis, and energy transference control. Given that two classes’ hazard theory is inadequate in hierarchical analysis of the second class of hazards, the hierarchy of causal influences model (Haslam et al., 2005) is considered to assist in modifying two classes’ hazard model. The proposed near-miss causation model in this study focuses on the transference of unrestrained energy from sources, through possible paths, and to potential receivers.

According to knowledge management, near-miss belongs to a type of tacit failure knowledge (Raviv et al., 2017; Zhou and Irizarry, 2016). The framework for near-miss management contains eight stages of discovery, report, identification, prioritization, causal analysis, solution, dissemination and evaluation. This systematic framework provides the feasibility of near-miss management on construction sites. Theoretical and practical contributions are twofold. As to theoretical contribution, corresponding findings provide a knowledge framework of near-miss information for construction safety researchers who can go on further study on near-miss management. As to practical contribution, the proposed framework is contributive to the guidance and encourage of near-miss practices on construction sites. However various obstacles will continue to hinder the utilization of near-miss management information for construction projects, as shown in Table 4. Some of the obstacles were found through the survey. Although corresponding suggestions were proposed in this study, obstacles cannot be overcome immediately. Because some obstacles are related to historical factors, company culture and individual habit, it will take a long time to overcome them (Cambraia et al., 2010; Raviv et al., 2017). Hence, the obstacles impeding near-miss management should be further studied and tackled. In addition, two more opportunities for future studies can be determined: (1) accident precursors can be explored from the near-miss database to support real-time early warning on construction project sites; and (2) near-misses can act as leading indicators to evaluate the safety performance of a construction project or a company for a certain period of time.

Aldred, R., Goodman, A. (2018). “Predictors of the frequency and subjective experience of cycling near misses: Findings from the first two years of the UK near miss project.” Accident Analysis and Preventi., 2018, 110, 161-170.

Baig, N., Wang, J., Elnahal, S., McNutt, T., Wright, J., DeWeese, T., and Terezakis, S. (2018). “Risk factors for near-miss events and safety incidents in pediatric radiation therapy.” Radiotherapy & Oncology Journal of the European Society for Therapeutic Radiology & Oncology, 127(2), 178-182.

Cambraia, F., Saurin, T., and Formoso, C. (2010). “Identification, analysis and dissemination of information on near misses: A case study in the construction industry.” Safety Sci., 2010, 48(1), 91-99.

Chen, B. (1996). Identification and assessment of hazards. Science and Technology Press of Sichuang Province, Chengdu, PRC.

Golzarpoor, B., Haas, C. T., Rayside, D., Kang, S., and Weston, M. (2018). “Improving construction industry process interoperability with industry foundation processes (IFP).” Adv. Eng. Inform., 38, 555-568.

Haslam, R. A., Hide, S. A., Gibb, A. G. F., Gyi, D. E., Pavitt, T., Atkinson, S., and Duff, A. R. (2005). “Contributing factors in construction accidents.” Appl. Ergon., 36(4), 401-415.

Kongsvik, T., Fenstad, J., and Wendelborg, C. (2012). “Between a rock and a hard place: Accident and near-miss reporting on offshore service vessels.” Safety Sci., 50(9), 1839-1846.

Maslow, A. H. (1943). “A theory of human motivation”. Psychol. Rev., 50(4), 370-96.

Maslow, A. H. (1954). Motivation and personality. Harper, New York, USA.

Oliveira, R., Zanella, A., and Camanho, S. (2018). “The assessment of corporate social responsibility: The construction of an industry ranking and identification of potential for improvement.” Eur. J. Oper. Res., In press.

Raviv, G., Shapira, A., and Fishbain, B. (2017). “AHP-based analysis of the risk potential of safety incidents: Case study of cranes in the construction industry.” Safety Sci., 91, 298-309.

Taylor, J. A., Lacovara, A. V., Smith, G. S., Pandian, R., and Lehto, M. (2014). “Near-miss narratives from the fire service: A Bayesian analysis.” Accident Analysis and Preventi., 62(1), 119-129.

Uth, H., and Wiese, N. (2004). “Central collecting and evaluating of major accidents and near-miss-events in the federal republic of Germany-results, experience, perspectives.” J. Hazard. Mater., 111, 139-145.

Wu, W., Yang, H., Chew, D. A. S., Yang, S., Gibb, A. G. F., and Li, Q. (2010b). “Towards an autonomous real-time tracking system of near-miss accidents on construction sites.” Automat. Constr., 19, 134-141.

Zhou, C., Ding, L., Skibniewski, M. J., Luo, H., and Jiang, S. (2017). “Characterizing time series of near-miss accidents in metro construction via complex network theory.” Safety Sci., 2017, 98, 145-158.

Zhou, Z., and Irizarry, J. (2016). “Integrated framework of modified accident energy release model and network theory to explore the full complexity of Hangzhou subway construction collapse.” J. Manage. Eng., 32(5), 05016013-1-9.

Comment 3: Conclusion: The author(s) stated that “Its main purpose is to explore potential use of near-miss information for promoting safety performance during the process of construction project management”. I expect the authors to shown this in the conclusion.  For instance was this achieved based on the evidence from the case study or not. This should be part of the conclusion.

Response:

According to the comment, a statement “A case study was implemented in the project of Nanjing Subway Line Four, in order to illustrate how to utilize near-miss as the effective information to deal with construction safety-related issues in a proactive way” was supplemented in the section of conclusions. In addition, it is also indicated that subway construction is only one part of the entire construction industry and there are some other sectors such as building construction, bridge construction, expressway construction, and dam construction. It is suggested that more case studies should be conducted in other types of construction projects for illustrating the general usability of near-miss information for construction safety.

Finally two editions of manuscript (one has track changes, and the other one is clean copy) are provided for your better reviewing.

Round  3

Reviewer 2 Report

The authors have addressed most of my concerns.